# Learning Identifiable Factorized Causal Representations of Cellular Responses

**Haiyi Mao**[*1,2]     **Romain Lopez**[1,3]     **Kai Liu**[1]     **Jan-Christian Huetter**[1]
**David Richmond**[1]     **Panayiotis V. Benos**[2,4]     **Lin Qiu**[†1]

[1]Genentech     [2]University of Pittsburgh     [3]Stanford University     [4]University of Florida

## Abstract

The study of cells and their responses to genetic or chemical perturbations promises to accelerate the discovery of therapeutic targets. However, designing adequate and insightful models for such data is difficult because the response of a cell to perturbations essentially depends on its biological context (e.g., genetic background or cell type). For example, while discovering therapeutic targets, one may want to enrich for drugs that specifically target a certain cell type. This challenge emphasizes the need for methods that explicitly take into account potential interactions between drugs and contexts. Towards this goal, we propose a novel Factorized Causal Representation (FCR) learning method that reveals causal structure in single-cell perturbation data from several cell lines. Based on the framework of identifiable deep generative models, FCR learns multiple cellular representations that are disentangled, comprised of covariate-specific ($\mathbf{z}_x$), treatment-specific ($\mathbf{z}_t$), and interaction-specific ($\mathbf{z}_{tx}$) blocks. Based on recent advances in non-linear ICA theory, we prove the component-wise identifiability of $\mathbf{z}_{tx}$ and block-wise identifiability of $\mathbf{z}_t$ and $\mathbf{z}_x$. Then, we present our implementation of FCR, and empirically demonstrate that it outperforms state-of-the-art baselines in various tasks across four single-cell datasets. The code is available on GitHub (`https://github.com/Genentech/fcr`).

## 1  Introduction

The recent experimental capabilities reached by single-cell perturbation technologies open up new opportunities for characterizing cellular behaviors (Dixit et al., 2016). For example, high-throughput screening of chemical and genetic perturbations highlighted vulnerabilities in numerous cancer cell lines (McFarland et al., 2020). Identifying these vulnerabilities is crucial for pinpointing therapeutic targets, facilitating drug discovery, and furthering our understanding of gene functions (Srivatsan et al., 2020).

The analysis of perturbation data involves modeling how cells respond to diverse treatments across biological contexts. This task is challenging for two main reasons. First, outcomes of perturbation experiments are quantified using single-cell RNA sequencing (scRNA-seq) technologies, whose measurements are noisy as well as high-dimensional (Grün et al., 2014). Second, cellular contexts are difficult to comprehensively model because they are extremely variable, encompassing cell types, tissue of origin, and genetic background (Wagner et al., 2016). This highlights the need to consider interaction effects between treatments and contextual covariates while modeling gene expression outcomes (Zapatero et al., 2023).

---

[*]This work was conducted during an internship at Genentech.
[†]Corresponding author. Email: lin.qiu.stats@gmail.com

38th Conference on Neural Information Processing Systems (NeurIPS 2024).

To address these challenges, several computational methods have been developed. Novel approaches based on causal representation learning provide better mechanistic interpretation of single-cell perturbation data (Lopez et al., 2023; Bereket and Karaletsos, 2023; Zhang et al., 2023). These methods belong to the family of identifiable deep generative models (Khemakhem et al., 2020; Lachapelle et al., 2022; Zheng et al., 2022) and therefore offer, to some extent, theoretical guarantees but remain limited to the analysis of data from a single cellular context. Another set of studies model cells across multiple contexts using latent linear additive models (Hetzel et al., 2022; Lotfollahi et al., 2023). However, due to their additive assumption, these models fail to characterize interactions between treatments and biological contexts.

To address these limitations, we introduce the Factorized Causal Representation (FCR) learning framework. This identifiable deep generative model learns representations of cells that take the form of three disentangled blocks, specific to treatments, biological contexts, and their interactions, respectively. We first present the proposed generative model and then provide a set of sufficient conditions for its identifiability, extending the work of Khemakhem et al. (2020) to the case of interacting covariates. We then present an implementation of our FCR method that builds upon the variational auto-encoder framework (Kingma and Welling, 2014) as well as adversarial regularization (Ganin et al., 2016). We demonstrate that FCR not only effectively identifies interactions but also surpasses state-of-the-art methods in various tasks across four single-cell datasets.

## 2 Related Work

**Learning Representations of Cellular Responses**   Learning representations of single-cell data is a powerful framework, with demonstrated impact in tasks such as imputation (Lopez et al., 2018), clustering (Trapnell et al., 2014; Zhu et al., 2019; Alquicira-Hernandez et al., 2019), and integration across modalities (Gayoso et al., 2022). Recent advancements have largely improved our capacity to predict cellular responses to drug treatments (Lotfollahi et al., 2019; Rampasek et al., 2019; Lotfollahi et al., 2021; Lopez et al., 2023; Bunne et al., 2023; Zapatero et al., 2023). Lotfollahi et al. (2023) and Hetzel et al. (2022) proposed generative models that additively combine treatment embeddings and biological context embeddings within a latent space. Wu et al. (2023) cast the prediction problem as a counterfactual inference problem. Despite these advancements, existing methods fail to address how treatments may preferentially impact specific cell types, a critical point for understanding the effects of drugs on biological systems.

**Identifiable non-linear Independent Component Analysis models**   A field where disentanglement is most important is non-linear Independent Component Analysis (non-linear ICA) (Hyvärinen and Pajunen, 1999), where information from a set of latent variables is mixed through a non-linear encoding function. It has long been known that such models (i.e., either the encoding function, or the sources) are non-identifiable without further assumption. However, some recent works (Hyvärinen et al., 2019; Lachapelle et al., 2022; Zheng et al., 2022) proved identifiability was possible in a non-stationary regime. Often, this assumption takes the form of conditional independence of the latent variables given some auxiliary (observed) variables. Notably, the iVAE framework from Khemakhem et al. (2020) offers disentanglement guarantees within this conditional VAE setup. Our work extends the theory of Khemakhem et al. (2020) to prove identifiability of the interaction terms between multiple such auxiliary variables.

This approach differs significantly from specific models in the Variational Autoencoder (VAE) literature (Kingma and Welling, 2014) such as the betaVAE (Higgins et al., 2016) and factorVAE (Kim and Mnih, 2018) that both address disentanglement learning. Indeed, these latter models lack theoretical foundation regarding the identifiability of their parameters or latent variables (Esmaeili et al., 2019; Chen and Grosse, 2018).

## 3 Preliminaries

Single-cell perturbation experiments characterize causes and effects at the cellular and molecular levels. Our objective is to disentangle the contributions of treatments, cellular covariates, and their interactions to model the effects of perturbations. Let $\mathbf{x} \in \mathcal{X} \subseteq \mathbb{R}^d$ be a vector of *covariates* representing intrinsic attributes of single cells, such as cell type, tissue of origin, or patient information.

Let $\mathbf{t} \in \mathcal{T} \subseteq \mathbb{R}^p$ represent the *treatment* or *intervention* applied to single cells and let $\mathbf{y} \in \mathcal{Y} \subseteq \mathbb{R}^k$ denote the gene expression levels (outcome).

## 3.1 Generative Model

We introduce a low-dimensional latent vector $\mathbf{z} \in \mathcal{Z} \subseteq \mathbb{R}^n$ that encodes cellular states after treatment $\mathbf{t}$ and in biological context $\mathbf{x}$. We assume a block structure for $\mathbf{z} = [\mathbf{z}_x, \mathbf{z}_{tx}, \mathbf{z}_t]$ with dimension $n = n_x + n_{tx} + n_t$. Here, $\mathbf{z}_x$ captures the effects of contextual covariates $\mathbf{x}$, $\mathbf{z}_t$ represents the direct effects of the treatment $\mathbf{t}$, and $\mathbf{z}_{tx}$ encodes the interaction effects between both terms.

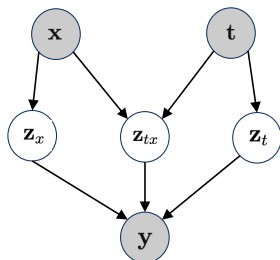

More precisely, the generative model is specified as follows. Latent representation $\mathbf{z}_x$ is generated from $\mathbf{x}$ according to distribution $\mathbf{z}_x \sim p_{\mathbf{z}_x|\mathbf{x}}(\mathbf{z}_x \mid \mathbf{x})$, $\mathbf{z}_t$ is generated from $\mathbf{t}$ following distribution $\mathbf{z}_t \sim p_{\mathbf{z}_t|\mathbf{t}}(\mathbf{z}_t \mid \mathbf{t})$, and $\mathbf{z}_{tx}$ from both $\mathbf{x}$ and $\mathbf{t}$ following distribution $\mathbf{z}_{tx} \sim p_{\mathbf{z}_{tx}|\mathbf{t},\mathbf{x}}(\mathbf{z}_{tx} \mid \mathbf{t}, \mathbf{x})$. The gene expression outcome vector $\mathbf{y}$ is then deterministically generated $\mathbf{y} = g(\mathbf{z}_x, \mathbf{z}_{tx}, \mathbf{z}_t)$,

Figure 1: Data generating process: shaded nodes denote observed variables. Empty nodes denote latent variables.

where $g$ is a smooth mixing function. A graphical representation of this generative model appears in Figure 1. For the control group (no treatment), the outcome is noted as $\mathbf{y}^0$, and the representation as $\mathbf{z}^0 = [\mathbf{z}_x^0, \mathbf{z}_{tx}^0, \mathbf{z}_t^0]$.

Learning each element in this triplet of latent variables is a sound approach for unravelling interaction effects. Indeed, $\mathbf{z}_x$ captures covariate-specific patterns that are invariant with respect to the perturbations, while also capturing the essential biological attributes tied to the cellular covariates. $\mathbf{z}_t$ captures the intrinsic effects of the treatments, irrespective of the covariates. $\mathbf{z}_{tx}$ unravels the interactions that a treatment could have with specific covariates. This last representation captures the nuanced manner in which distinct cell types, tissues, or patient groups react to treatments, reflecting the diversity and specificity of biological responses.

We note that our model does not take into account observation noise, since $g$ is a *deterministic* function in our assumption. However, our theory may be readily extended to incorporate Gaussian observation noise (Khemakhem et al., 2020), as well as counting observation noise (Lopez et al., 2024).

## 3.2 Model Identifiability

We now introduce the definitions for the different classes of disentangled models that will appear later in the manuscript. Analogous definitions appear in previous work from Von Kügelgen et al. (2021) and Kong et al. (2022). Throughout this section, $\mathbf{z} \in \mathcal{Z}$ denotes a (random) latent vector and $\mathbf{y} \in \mathcal{Y}$ denotes an observed vector. $g : \mathcal{Z} \rightarrow \mathcal{Y}$ is an unknown mixing function such that $\mathbf{y} = g(\mathbf{z})$.

**Definition 3.1** (*Component-wise Identifiability*). We say that latent vector $\mathbf{z}$ is *identifiable* from data $\mathbf{y}$ if for any other latent vector $\hat{\mathbf{z}}$ such that $g(\hat{\mathbf{z}})$ and $g(\mathbf{z})$ are equal in distribution, $\mathbf{z}$ and $\hat{\mathbf{z}}$ are equal up to a permutation of the indices, and deformation of each component by invertible scalar functions. More precisely, there exists a permutation $\pi$ of $\{1, \ldots, n\}$, and invertible scalar functions $h_j$ such that for all $j \in \{1, \ldots, n\}$:

$$\hat{z}_j = h_j(z_{\pi(j)}), \tag{1}$$

where $z_j$ and $\hat{z}_{\pi(j)}$ are the $\pi(j)$-th components of $\mathbf{z}$ and $\hat{\mathbf{z}}$ respectively.

**Definition 3.2** (*Block-wise Identifiability*). For $1 \leq n_1 < n_2 \leq n$, we denote as $\mathbf{z}_{[n_1:n_2]} \in \mathbb{R}^{n_2-n_1+1}$ the block of $\mathbf{z}$ from index $n_1$ to $n_2$. We say that latent vector $\mathbf{z}_{[n_1:n_2]}$ is *block-identifiable* from data $\mathbf{y}$ if for any other latent vector $\hat{\mathbf{z}}$ such that $g(\hat{\mathbf{z}})$ and $g(\mathbf{z})$ are equal in distribution, $\mathbf{z}_{[n_1:n_2]}$ and $\hat{\mathbf{z}}_{[n_1:n_2]}$ are equal up to an invertible function $h$:

$$\hat{\mathbf{z}}_{[n_1:n_2]} = h(\mathbf{z}_{[n_1:n_2]}), \tag{2}$$

where $\hat{\mathbf{z}}_{[n_1:n_2]}$ is the corresponding block in the estimated vector $\hat{\mathbf{z}}$.

# 4 Identifiability Results

One strong advantage of the FCR framework is that it comes with strong theoretical guarantees concerning the disentanglement of its factorized latent space. We first prove the *component-wise identifiability* of the interaction variable $\mathbf{z}_{tx}$ under the assumption of sufficient experimental variability (Khemakhem et al., 2020) (Section 4.1). Then, we demonstrate the block-identifiability of $\mathbf{z}_t$ and $\mathbf{z}_x$ by exploiting their invariance with respect to $\mathbf{x}$ and $\mathbf{t}$, respectively (Section 4.2). These guarantees are important, as they ensure that the obtained latent variables will have desirable semantics. For example, given our theoretical results, the obtained interaction embedding $\hat{\mathbf{z}}_{tx}$ must be reflective of the ground-truth interactions $\mathbf{z}_{tx}$ only, and not of any of the other latent variables.

## 4.1 Component-wise identifiability of $\mathbf{z}_{tx}$

Our proof relies on three technical assumptions. Two are classical from the nonlinear ICA literature, and the last one relates to the observability of a complementary set of experiments for identifiability of interactions.

**Assumption 4.1.** The probability density function of the prior distribution for the latent variables is smooth and positive, i.e. $p_{\mathbf{z}|\mathbf{t},\mathbf{x}}(\mathbf{z} \mid \mathbf{t}, \mathbf{x}) > 0$ for all $(\mathbf{z}, \mathbf{t}, \mathbf{x}) \in \mathcal{Z} \times \mathcal{T} \times \mathcal{X}$.

**Assumption 4.2.** The components of $\mathbf{z}$ are conditionally independent given $\mathbf{t}$ and $\mathbf{x}$.

**Assumption 4.3.** (*Experimental Sufficiency*) There exist at least $2n_{tx} + 1$ distinct values for the vector $[\mathbf{t}, \mathbf{x}]$ in the experimental design. One such setting can be referred to as a control condition and is noted as $[\mathbf{t}_0, \mathbf{x}_0]$. Additionally, for any non-control environment $(\mathbf{t}_i, \mathbf{x}_i)$ for $i \in \{1, \ldots, 2n_{tx}\}$, we assume there always exist corresponding switched experiments under the settings $(\mathbf{t}_0, \mathbf{x}_i)$ and $(\mathbf{t}_i, \mathbf{x}_0)$.

Assumption 4.3 is novel and essentially mandates that we conduct a sufficient number of experiments with cross-referenced covariates and treatments. This ensures that we can observe the specific drug response related to each covariate, and is often how such experiments are designed in practice.

**Theorem 4.4.** *Let us first define* $\mathbf{v}(\mathbf{z}_{tx}, \mathbf{t}, \mathbf{x})$ *as the vector:*

$$\mathbf{v}(\mathbf{z}_{tx}, \mathbf{t}, \mathbf{x}) = \left[ \frac{\partial q_{n_x+1}(z_{n_x+1}, \mathbf{t}, \mathbf{x})}{\partial z_{n_x+1}}, \cdots, \frac{\partial q_{n_x+n_{tx}}(z_{n_x+n_{tx}}, \mathbf{t}, \mathbf{x})}{\partial z_{n_x+n_{tx}}}, \right.$$
$$\left. \frac{\partial^2 q_{n_x+1}(z_{n_x+1}, \mathbf{t}, \mathbf{x})}{\partial z_{n_x+1}^2}, \cdots, \frac{\partial^2 q_{n_x+n_{tx}}(z_{n_x+n_{tx}}, \mathbf{t}, \mathbf{x})}{\partial z_{n_x+n_{tx}}^2} \right],$$

*where* $q_i$ *denotes the logarithm of probability density* $p_{\mathbf{z}_i|\mathbf{t},\mathbf{x}}$ *for component* $\mathbf{z}_i$. *If in addition to the assumptions 4.1, 4.2, 4.3, we assume that the* $2n_{tx}$ *vectors*

$$\{\mathbf{v}(\mathbf{z}_{tx}, \mathbf{t}_i, \mathbf{x}_i) + \mathbf{v}(\mathbf{z}_{tx}, \mathbf{t}_0, \mathbf{x}_0) - \mathbf{v}(\mathbf{z}_{tx}, \mathbf{t}_0, \mathbf{x}_i) - \mathbf{v}(\mathbf{z}_{tx}, \mathbf{t}_i, \mathbf{x}_0)\}_{i=1}^{2n_{tx}}, \tag{3}$$

*are linearly independent then* $\mathbf{z}_{tx}$ *is component-wise identifiable.*

The proof appears in Appendix A. Theorem 4.4 extends the concept of linear independence found in nonlinear ICA (Khemakhem et al., 2020). Unlike the original theory, where auxiliary variables must induce sufficient variations of the latent variables, we are confronted with a case where treatments and contexts must have sufficient variability in combination. For example, when $\mathbf{v}$ is linear with respect to both $\mathbf{t}$ and $\mathbf{x}$ the vector of interest becomes the null vector. In this trivial case, the theorem's assumption is never satisfied ($z_{tx}$ indeed has no purpose in that particular model). However, under a rich class of model with complex interaction patterns, our model will be able to infer informative latent variables.

## 4.2 Block-wise identifiability of $\mathbf{z}_x$ and $\mathbf{z}_t$

To prove the block-wise identifiability of $\mathbf{z}_x$ and $\mathbf{z}_t$, we exploit their invariance properties: $\mathbf{z}_t$ remains unchanged across different values of $\mathbf{x}$, while $\mathbf{z}_x$ is stable across variations in $\mathbf{t}$. This invariance allows us to distinguish these blocks from the interaction terms $\mathbf{z}_{tx}$. Additionally, because this invariance reflects latent features' stability despite perturbations, its utilization can enable deeper biological insights and interpretability. For instance, $\mathbf{z}_x$ might represent stable cellular characteristics

that persist across different treatments, while $\mathbf{z}_t$ could capture consistent treatment effects across various cell types.

We now state our main identifiability result for $\mathbf{z}_x$ and $\mathbf{z}_t$.

**Theorem 4.5.** *We follow Assumptions 4.1, 4.2, 4.3, and the one from Theorem 4.4. We note as $\mathcal{S}(Z)$ the set of subsets $S \subseteq \mathcal{Z}$ of $\mathcal{Z}$ that satisfy the following two conditions:*

*(i) $S$ has nonzero probability measure, i.e. $\mathbb{P}(\mathbf{z} \in S \mid \mathbf{t} = \mathbf{t}', \mathbf{x} = \mathbf{x}') > 0$ for any $\mathbf{t}' \in \mathcal{T}$ and $\mathbf{x}' \in \mathcal{X}$.*

*(ii) $S$ cannot be expressed as $A_{\mathbf{z}_x} \times \mathcal{Z}_{tx} \times \mathcal{Z}_t$ for any $A_{\mathbf{z}_x} \subset \mathcal{Z}_x$ or as $\mathcal{Z}_x \times \mathcal{Z}_{tx} \times A_{\mathbf{z}_t}$ for any $A_{\mathbf{z}_t} \subset \mathcal{Z}_t$.*

*We have the following identifiability result. If for all $S \in \mathcal{S}(\mathcal{Z})$, there exists $(\mathbf{t}_1, \mathbf{t}_2) \in \mathcal{T} \times \mathcal{T}$ and $\mathbf{x} \in \mathcal{X}$ such that*

$$\int_{\mathbf{z} \in S} p_{\mathbf{z}|\mathbf{t},\mathbf{x}}(\mathbf{z} \mid \mathbf{t}_1, \mathbf{x}) d\mathbf{z} \neq \int_{\mathbf{z} \in S} p_{\mathbf{z}|\mathbf{t},\mathbf{x}}(\mathbf{z} \mid \mathbf{t}_2, \mathbf{x}) d\mathbf{z}, \tag{4}$$

*and there also exists $(\mathbf{x}_1, \mathbf{x}_2) \in \mathcal{X} \times \mathcal{X}$ and $\mathbf{t} \in \mathcal{T}$ such that*

$$\int_{\mathbf{z} \in S} p_{\mathbf{z}|\mathbf{t},\mathbf{x}}(\mathbf{z} \mid \mathbf{t}, \mathbf{x}_1) d\mathbf{z} \neq \int_{\mathbf{z} \in S} p_{\mathbf{z}|\mathbf{t},\mathbf{x}}(\mathbf{z} \mid \mathbf{t}, \mathbf{x}_2) d\mathbf{z}, \tag{5}$$

*then $\mathbf{z}_t$ and $\mathbf{z}_x$ are block-wise identifiable.*

The proof appears in Appendix A, and is adapted from Kong et al. (2022). Our main assumption is that the conditional distribution $p_{\mathbf{z}|\mathbf{t},\mathbf{x}}(\mathbf{z} \mid \mathbf{t}, \mathbf{x})$ undergoes substantive changes when spanning different treatments $\mathbf{t}$ and covariates $\mathbf{x}$. When treatments differ markedly from each other in their mechanisms and effects, the probability distributions of the latent variables conditioned on these treatments are unlikely to be identical.

## 5 Methodology

We now propose a tangible implementation of our method, termed Factorized Causal Representation (FCR) learning. Our approach has three components:

1. A variational inference approach to estimate the representations from our FCR model. Our model and inference architecture is specifically designed to learn disentangled representations $\mathbf{z}_x$, $\mathbf{z}_{tx}$, $\mathbf{z}_t$ (Section 5.1).

2. A regularization method that enforces independence between $\mathbf{z}_x$ and $\mathbf{t}$, and encourages variability of $\mathbf{z}_t$ with respect to $\mathbf{x}$ (Section 5.2).

3. A second regularization technique to ensure that the conditional independence properties $\mathbf{z}_x \perp\!\!\!\perp \mathbf{z}_{tx} \mid \mathbf{x}$ and $\mathbf{z}_t \perp\!\!\!\perp \mathbf{z}_{tx} \mid \mathbf{t}$ are satisfied (Section 5.3).

The main computational structure of the model is illustrated as a schematic in Figure 2.

### 5.1 Model Specification and Variational Inference

**Model Specification**  We parameterize the probability distributions as follows:

$$p(\mathbf{z}_x \mid \mathbf{x}) \coloneqq \text{Normal}(f_\mu^x(\mathbf{x}), f_\sigma^x(\mathbf{x})) \tag{6}$$

$$p(\mathbf{z}_t \mid \mathbf{t}) \coloneqq \text{Normal}(f_\mu^t(\mathbf{t}), f_\sigma^t(\mathbf{t})) \tag{7}$$

$$p(\mathbf{z}_{tx} \mid \mathbf{t}, \mathbf{x}) \coloneqq \text{Normal}(f_\mu^{t,x}(\mathbf{t}, \mathbf{x}), f_\sigma^{t,x}(\mathbf{t}, \mathbf{x})), \tag{8}$$

where all the above functions are parameterized by neural networks.

To prevent $\mathbf{z}_{tx}$ from having trivial dependencies with respect to $\mathbf{t}$ and $\mathbf{x}$, we explicitly encourage its prior to capture interactions between $\mathbf{x}$ and $\mathbf{t}$ by designing the functions $f_\mu^{t,x}$ and $f_\sigma^{t,x}$ to be of the form:

$$f_\mu^{t,x} = f_\mu(k_\mathbf{x}(\mathbf{x}) \odot k_\mathbf{t}(\mathbf{t})) \tag{9}$$

$$f_\sigma^{t,x} = f_\sigma(k_\mathbf{x}(\mathbf{x}) \odot k_\mathbf{t}(\mathbf{t})), \tag{10}$$

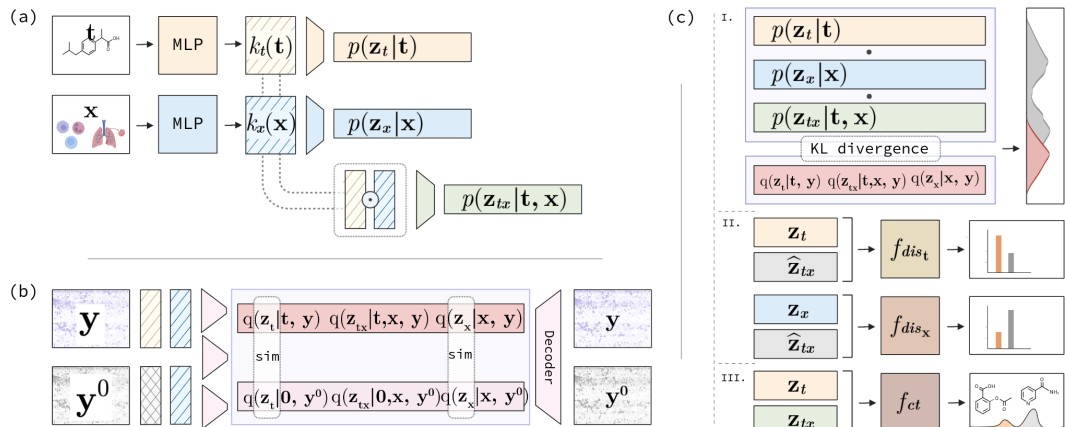

Figure 2: The illustration of FCR models. (a) is the component for $p(\mathbf{z}_x \mid \mathbf{x}), p(\mathbf{z}_t \mid \mathbf{t})$ and $p(\mathbf{z}_{tx} \mid \mathbf{t}, \mathbf{x})$ (b) is the component to estimate $q(\mathbf{z}_x \mid \mathbf{x}, \mathbf{y}), q(\mathbf{z}_t \mid \mathbf{t}, \mathbf{y})$ and $q(\mathbf{z}_{t,x} \mid \mathbf{t}, \mathbf{x}, \mathbf{y})$. Note that $\mathbf{0}$ indicates $\mathbf{t} = \mathbf{0}$ representing the control samples. (c) computational diagrams to estimate the Kullback-Leibler divergences, causal structure regularization and permutation discriminators.

where $k_x(\mathbf{x})$ and $k_t(\mathbf{t})$ represent the embeddings for the cellular covariates and treatments, respectively, while $\odot$ denotes the Hadamard product.

**Variational Inference** Because the posterior distribution of the latent variables are intractable, we use the variational autoencoder framework (Kingma and Welling, 2014) to jointly learn the model's parameters and an approximation to the posterior, following the approach used in previous causal representation learning work (Khemakhem et al., 2020). We consider the following mean-field variational approximation to the posterior distribution:

$$q_\phi(\mathbf{z}_x, \mathbf{z}_t, \mathbf{z}_{tx} \mid \mathbf{x}, \mathbf{t}, \mathbf{y}) = q_\phi(\mathbf{z}_x \mid \mathbf{x}, \mathbf{y}) q_\phi(\mathbf{z}_t \mid \mathbf{t}, \mathbf{y}) q_\phi(\mathbf{z}_{tx} \mid \mathbf{t}, \mathbf{x}, \mathbf{y}). \tag{11}$$

Following the graphical model from Figure 1, the Evidence Lower Bound (ELBO) is derived as:

$$\begin{aligned}
\mathcal{L}_{\text{ELBO}} =\ & \mathbb{E}_{q_\phi(\mathbf{z}_x, \mathbf{z}_t, \mathbf{z}_{tx} \mid \mathbf{x}, \mathbf{t}, \mathbf{y})} \log p_\theta(\mathbf{y} \mid \mathbf{z}_x, \mathbf{z}_t, \mathbf{z}_{tx}) \\
& - D_{KL}(q_\phi(\mathbf{z}_x \mid \mathbf{x}, \mathbf{y}) \| p_\theta(\mathbf{z}_x \mid \mathbf{x})) \\
& - D_{KL}(q_\phi(\mathbf{z}_t \mid \mathbf{t}, \mathbf{y}) \| p_\theta(\mathbf{z}_t \mid \mathbf{t})) \\
& - D_{KL}(q_\phi(\mathbf{z}_{tx} \mid \mathbf{t}, \mathbf{x}, \mathbf{y}) \| p_\theta(\mathbf{z}_{tx} \mid \mathbf{t}, \mathbf{x})),
\end{aligned} \tag{12}$$

where $\theta$ and $\phi$ denote the parameters of the generative model and the inference networks, respectively. $D_{KL}$ denotes the Kullback–Leibler divergence between two probability distributions. For simplicity, we omit $\phi$ and $\theta$ as well as script notations in the following sections, wherever appropriate. The derivation of the ELBO appears in Appendix B.

The variational distributions defined in Equation 11 are parameterized as follows:

$$q(\mathbf{z}_x \mid \mathbf{x}, \mathbf{y}) \coloneqq \text{Normal}(g_\mu^x(\mathbf{x}, \mathbf{y}), g_\sigma^x(\mathbf{x}, \mathbf{y})) \tag{13}$$

$$q(\mathbf{z}_t \mid \mathbf{t}, \mathbf{y}) \coloneqq \text{Normal}(g_\mu^t(\mathbf{t}, \mathbf{y}), g_\sigma^t(\mathbf{t}, \mathbf{y})) \tag{14}$$

$$q(\mathbf{z}_{tx} \mid \mathbf{t}, \mathbf{x}, \mathbf{y}) \coloneqq \text{Normal}(g_\mu^{t,x}(\mathbf{t}, \mathbf{x}, \mathbf{y}), g_\sigma^{t,x}(\mathbf{t}, \mathbf{x}, \mathbf{y})), \tag{15}$$

where all the functions introduced above are parameterized by neural networks.

### 5.2 Causal Structure Regularization

We exploit both the variability of $\mathbf{z}_t$ and the invariance of $\mathbf{z}_x$ when comparing control and treated groups that share the same covariates. Specifically, our goal is to enforce the resemblance between $\mathbf{z}_x$ and $\mathbf{z}_x^0$ while reducing the congruence of $\mathbf{z}_t$ and $\mathbf{z}_t^0$. Towards this end, we first add the following score as a regularizer,

$$\mathcal{L}_{\text{sim}} = \mathbb{E}_{q(\mathbf{z}_x, \mathbf{z}_t \mid \mathbf{x}, \mathbf{t}) q(\mathbf{z}_x^0, \mathbf{z}_t^0 \mid \mathbf{x}_0, \mathbf{t}_0)} \left[ \text{sim}\left(\mathbf{z}_t, \mathbf{z}_t^0\right) - \text{sim}\left(\mathbf{z}_x, \mathbf{z}_x^0\right) \right], \tag{16}$$

where $\text{sim}(\cdot)$ denotes the cosine similarity. Second, we introduce a classifier $f_{\text{ct}}$ to predict the treatments $\mathbf{t}$ from $[\mathbf{z}_t, \mathbf{z}_{tx}]$ and $[\mathbf{z}_t^0, \mathbf{z}_{tx}^0]$ as follows,

$$\tilde{\mathbf{t}} = f_{\text{ct}}\left([\mathbf{z}_t, \mathbf{z}_{tx}], [\mathbf{z}_t^0, \mathbf{z}_{tx}^0]\right). \tag{17}$$

The predicted treatment probability vector $\tilde{\mathbf{t}}$ is then used for the computation of a cross-entropy loss

$$\mathcal{L}_{\text{ct}} = -\mathbb{E}_{t, q(\mathbf{z}_{tx}, \mathbf{z}_t | \mathbf{x}, \mathbf{t}) q(\mathbf{z}_{tx}^0, \mathbf{z}_t^0 | \mathbf{x}^0, \mathbf{t}^0)}[\mathbf{t} \cdot \log(\tilde{\mathbf{t}})]. \tag{18}$$

### 5.3 Permutation Discriminators

We want to ensure the conditions of Assumption 4.2 and Theorem 4.4, specifically that $\mathbf{z}_{tx} \perp\!\!\!\perp \mathbf{z}_t \mid \mathbf{t}$ and $\mathbf{z}_{tx} \perp\!\!\!\perp \mathbf{z}_x \mid \mathbf{x}$. Towards this goal, we use the following proposition, establishing a connection between exchangeability and conditional independence.

**Proposition 5.1.** *(Bellot and van der Schaar, 2019) Let $X, Y$ and $Z$ be three random variables. Under the assumption of $X \perp\!\!\!\perp Y \mid Z$, we have the samples $(X_i, Y_i, Z_i)_{i=1}^M$ and permuted samples $(X_{\pi(i)}, Y_i, Z_i)_{i=1}^M$ with a permutation function $\pi$. The corresponding statistics $\rho_i$ of $(X_i, Y_i)_{i=1}^M$ and $\rho_{\pi(i)}$ of $(X_{\pi(i)}, Y_i)_{i=1}^M$ are exchangeable.*

This proposition states that permutation will not change the independence between two conditionally independent random variables. We therefore propose to use permutation discriminators for $\mathbf{z}_x, \mathbf{z}_{tx}$ and $\mathbf{z}_t, \mathbf{z}_{tx}$. First, we initially permute $\mathbf{z}_{tx}$ within the triplet $(\mathbf{z}_x^{(j)}, \mathbf{z}_{tx}^{(j)}, \mathbf{x}^{(j)} = \mathbf{x}_i)_{j=1}^M$ to yield $(\mathbf{z}_x^{(j)}, \mathbf{z}_{tx}^{\pi(j)}, \mathbf{x}^{(j)} = \mathbf{x}_i)_{j=1}^M$. Then, we train a binary classifier (the discriminator) to predict whether samples have been permuted or not. We denote the permutation label as $l$ and predict it as

$$\tilde{l} = f_{\text{dis}_\mathbf{x}}(\mathbf{z}_x, \hat{\mathbf{z}}_{tx}, \mathbf{x}), \tag{19}$$

where $\hat{\mathbf{z}}_{tx}$ could be permuted or non-permuted $\mathbf{z}_{tx}$ samples. If $\mathbf{z}_x$ and $\mathbf{z}_{tx}$ are indeed independent given $\mathbf{x}$, the discriminator should be unable to distinguish between the permuted and the original samples. For each discriminator, we add a regularization term that consists of the cross-entropy loss

$$\mathcal{L}_{\text{dis}_\mathbf{x}} = -\mathbb{E}_{\mathbf{x}, q(\mathbf{z}_{tx}, \mathbf{z}_x | \mathbf{x}, \mathbf{t})}[l \log(\tilde{l})]. \tag{20}$$

We proceed similarly to make sure that $\mathbf{z}_t$ and $\mathbf{z}_{tx}$ are independent conditionally on $\mathbf{t}$.

### 5.4 Objective function

Finally, we specify the overall loss for our model as

$$\mathcal{L}_{\text{total}} = \mathcal{L}_{\text{ELBO}} + \omega_1 \mathcal{L}_{\text{sim}} + \omega_2 \mathcal{L}_{\text{ct}} - \omega_3(\mathcal{L}_{\text{dis}_\mathbf{x}} + \mathcal{L}_{\text{dis}_\mathbf{t}}), \tag{21}$$

where $\omega_1, \omega_2, \omega_3 > 0$ are hyperparameters. To concurrently train both the representations and the discriminators, we employ an adversarial training approach as follows,

$$\max_{f_{\text{dis}_\mathbf{t}}, f_{\text{dis}_\mathbf{x}}} \min_{\theta, \phi, f_{\text{ct}}} \mathcal{L}_{\text{total}}. \tag{22}$$

The training procedure and the procedure used for hyperparameters selection are detailed in Appendix C and D, respectively.

## 6 Experiments

In this section, we are pursuing three primary objectives. First, we seek to validate the FCR method's proficiency in capturing the designated causal structure within the latent space through both clustering analysis (Section 6.1) and statistical testing of independence (Section 6.2), respectively. Second, we evaluate the method's efficacy in predict single-cell level conditional cellular responses (Section 6.3). We note that the current implementation of FCR does not make use of general embeddings for $\mathbf{t}$ or $\mathbf{x}$, and for that reason we do not perform experiments to predict cellular responses to unseen treatments and cell types (covariates).

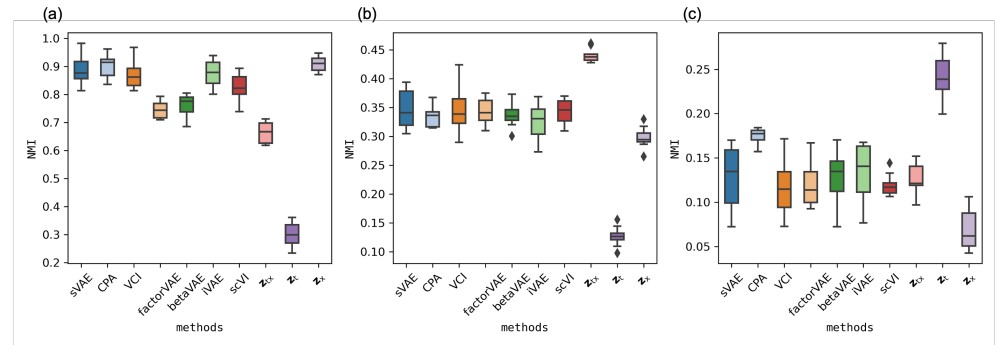

Figure 3: Clustering results for the sciPlex dataset. Normalized Mutual Information (NMI) values for clustering based on: (a) covariates $\mathbf{x}$; (b) combined covariates and treatments $[\mathbf{x}; \mathbf{t}]$; (c) treatments $\mathbf{t}$.

**Datasets** To evaluate the efficacy and robustness of the FCR method, we conducted our study on four real single-cell perturbation datasets (Appendix E). The first of these is the sciPlex dataset (Srivatsan et al., 2020), which provides insights into the impact of several HDAC (Histone Deacetylase) inhibitors on a total of 11,755 cells from three distinct cell lines: A549, K562, and MCF7. Each of these cell lines was subjected to treatment in two independent replicate experiments, using five different drug dosages: 0 nM (control), 10 nM, 100 nM, 1 μM, and 10 μM. The subsequent three datasets are sourced from (McFarland et al., 2020), which executed several large-scale experiments in varied settings. The multiPlex-Tram dataset contains 13,713 cells from 24 cell lines, treated with Trametinib and a DMSO control over durations of 3, 6, 12, 24, and 48 hours. The multiPlex-7 dataset spans 61,552 cells across 97 cell lines, subjected to seven different treatments. Finally, the multiPlex-9 dataset incorporates 19,524 cells from 24 cell lines, undergoing a series of nine treatments.

**Baselines** We benchmarked our method against several established representation learning methods: (1) scVI (Lopez et al., 2018), (2) iVAE (Khemakhem et al., 2020), (3) $\beta$VAE (Higgins et al., 2016), (4) factorVAE (Kim and Mnih, 2018), (5) VCI (Wu et al., 2023), (6) CPA (Lotfollahi et al., 2023), (7) scGEN (Lotfollahi et al., 2019) (8) sVAE (Lopez et al., 2023), (9) CINEMA-OT (Dong et al., 2023). For the clustering analysis, we employed all the inferred latent variables from each baseline method. For the conditional independence test, we selected a random subset of latent variables for each baseline matching the number of latent variables of FCR. We then tested each subset and repeated the process a total of ten times for each baseline using different random subsets to yield the best results. Specifically, CINEMA-OT and scGEN address only binary treatments/perturbations, so they are only considered in the cellular response predictions tasks.

Results on additional dataset, simulation studies, ablation studies, data visualization, and biological interpretation of the latent variables appear in Appendix F.

### 6.1 Clustering Analysis on Covariates, Treatments, and Combined Features

We evaluated the performance of the obtained latent representations in capturing three key aspects: the cellular covariates $\mathbf{x}$, the treatments $\mathbf{t}$, and their interaction $(\mathbf{x}, \mathbf{t})$. To assess each latent representation, we applied clustering and compared the fidelity of the resulting cluster labels with the corresponding $\mathbf{x}$, $\mathbf{t}$, or $(\mathbf{x}, \mathbf{t})$ from the original data. This approach allowed us to gauge how well the latent representations preserved the underlying structure of the cellular covariates, treatments, and their interactions. We performed clustering using the Leiden algorithm (Traag et al., 2019). To assess the fidelity between two sets of cluster labels, we employed the Normalized Mutual Information (NMI) metric (Kim et al., 2019). Higher NMI values indicate better alignment between the clustering results and the original data structure. We conducted this clustering and fidelity assessment on our model's latent variables $\mathbf{z}_x$, $\mathbf{z}_{tx}$, and $\mathbf{z}_t$, as well as on the variables obtained from baseline methods.

Our results highlight that $\mathbf{z}_x$ has superior performance in clustering on covariates $\mathbf{x}$ compared to all other available latent representations (Figure 3). Similarly, $\mathbf{z}_t$ performs best for clustering on treatments $\mathbf{t}$. Finally, $\mathbf{z}_{tx}$ outperforms all other methods when clustering jointly on $\mathbf{t}$ and $\mathbf{x}$, showing that it faithfully represents the combined features of both $\mathbf{x}$ and $\mathbf{t}$. Specifically, for the sciPlex datasets,

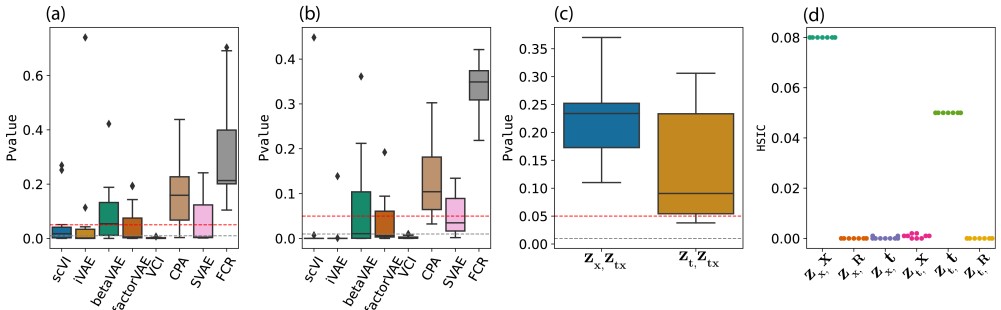

Figure 4: Statistical Conditional Independence Testing Results (a) p-values for the conditional independence test of $\mathbf{z}_x \perp\!\!\!\perp \mathbf{t} \mid \mathbf{x}$. The red dashed line indicates the 0.05 level. (b) p-values for the conditional independence test of $\mathbf{z}_t \perp\!\!\!\perp \mathbf{x} \mid \mathbf{t}$. (c) p-values for the conditional independence tests $\mathbf{z}_x \perp\!\!\!\perp \mathbf{z}_{tx} \mid \mathbf{x}$ and $\mathbf{z}_t \perp\!\!\!\perp \mathbf{z}_{tx} \mid \mathbf{t}$. (d) HSIC values for assessment of marginal independence of $\mathbf{z}_x$ with $\mathbf{x}$, $\mathbf{t}$ and random numbers ($\mathbf{R}$); as well as $\mathbf{z}_t$ with $\mathbf{x}$, $\mathbf{t}$ and random numbers ($\mathbf{R}$).

clustering on $\mathbf{x}$ yields better results than clustering based on both $(\mathbf{t}, \mathbf{x})$ or on solely $\mathbf{t}$. This can be attributed to the HDAC inhibitors exhibiting minimal impact on distinct cell lines until a maximal concentration of 10 μM was reached. We report similar results for the other datasets in Appendix F. Taken together, these results suggest that FCR effectively learns disentangled representations across different datasets.

## 6.2 Statistical Conditional Independence Testing

We validated the disentanglement of our latent representations via conditional independence testing, implemented as the Kernel Conditional Independence (KCI) (Zhang et al., 2012) tests. Our investigations focused on the following relationships: (1) $\mathbf{z}_x \perp\!\!\!\perp \mathbf{t} \mid \mathbf{x}$; (2) $\mathbf{z}_t \perp\!\!\!\perp \mathbf{x} \mid \mathbf{t}$; (3) $\mathbf{z}_t \perp\!\!\!\perp \mathbf{z}_{tx} \mid \mathbf{t}$; (4) $\mathbf{z}_x \perp\!\!\!\perp \mathbf{z}_{tx} \mid \mathbf{x}$. In conjunction with these tests, we also employed the Hilbert-Schmidt Independence Criterion (HSIC) (Gretton et al., 2005) to evaluate the (marginal) independence between: (a) $\mathbf{z}_t$ and $\mathbf{t}$; (b) $\mathbf{z}_x$ and $\mathbf{x}$. Our aim was two-fold. First, we wanted to assess whether the factors in our latent space complied with the necessary conditional independence statements, corroborating a well-captured causal structure. Second, we wanted to determine the dependence between our latent representations and their respective observed variables.

We focus our presentation of the results on the sciPlex dataset in the main text (results for the other datasets imply similar conclusions and appear in Appendix F.4). We first examined the results of testing for conditional independence statements $\mathbf{z}_x \perp\!\!\!\perp \mathbf{t} \mid \mathbf{x}$ and $\mathbf{z}_t \perp\!\!\!\perp \mathbf{x} \mid \mathbf{t}$ (Figure 4ab). In this experiment, all the baseline methods produced p-values smaller than 0.05. This implies a rejection of the null hypothesis (conditional independence) for the baseline methods, and suggests that their representations failed to maintain the desired conditional independence statements. Second, we examined the results of testing for conditional independence statements $\mathbf{z}_x \perp\!\!\!\perp \mathbf{z}_{tx} \mid \mathbf{x}$ and $\mathbf{z}_t \perp\!\!\!\perp \mathbf{z}_{tx} \mid \mathbf{t}$ (Figure 4c). Interestingly, this assessment also quantifies the efficacy of our permutation discriminators. The observed p-values generally exceed 0.05, suggesting that the null hypothesis cannot be rejected. Finally, we use the HSIC to report estimates of mutual information (assessing for marginal independence). Low HSIC values suggest poor dependence between the pairs of random variables. We report the HSIC values for assessing $\mathbf{z}_x \perp\!\!\!\perp \mathbf{x}$, $\mathbf{z}_x \perp\!\!\!\perp \mathbf{t}$, $\mathbf{z}_x \perp\!\!\!\perp \mathbf{R}$, as well as $\mathbf{z}_t \perp\!\!\!\perp \mathbf{x}$, $\mathbf{z}_t \perp\!\!\!\perp \mathbf{t}$, and $\mathbf{z}_t \perp\!\!\!\perp \mathbf{R}$, where $\mathbf{R}$ represents simulated random vectors (Figure 4d). Contrasting our representations with randomly simulated vectors, we observe that both $\mathbf{z}_x$ and $\mathbf{x}$, as well as $\mathbf{z}_t$ and $\mathbf{t}$, have HSIC values far from zero, indicating a high dependence. The contrast in results between our approach and the baseline methods highlights FCR's nuanced capability in capturing and preserving causal structures.

## 6.3 Conditional Cellular Responses Prediction

Our analysis demonstrates that treatments often elicit covariate (cell line) specific responses. Consequently, accurately predicting outcomes for novel drugs or cell lines becomes challenging without a careful consideration of the similarity in $\mathbf{z}_{tx}$ and how $\mathbf{t}$ interacts with $\mathbf{x}$. Unlike previous literature,

| Datasets | Methods | | | | | |
| --- | --- | --- | --- | --- | --- | --- |
| | FCR (ours) | VCI | CPA | scGEN | sVAE | CINEMA-OT |
| sciPlex | $\mathbf{0.87}_{\pm 0.02}$ | $0.86_{\pm 0.03}$ | $0.86_{\pm 0.03}$ | $0.56_{\pm 0.05}$ | $0.84_{\pm 0.02}$ | $0.51_{\pm 0.08}$ |
| multiPlex-Tram | $\mathbf{0.90}_{\pm 0.03}$ | $0.89_{\pm 0.04}$ | $0.88_{\pm 0.04}$ | $0.52_{\pm 0.06}$ | $0.87_{\pm 0.02}$ | $0.33_{\pm 0.09}$ |
| multiPlex-7 | $\mathbf{0.83}_{\pm 0.03}$ | $0.81_{\pm 0.04}$ | $0.80_{\pm 0.04}$ | $0.49_{\pm 0.11}$ | $0.77_{\pm 0.02}$ | $0.41_{\pm 0.09}$ |
| multiPlex-9 | $0.78_{\pm 0.03}$ | $0.78_{\pm 0.04}$ | $\mathbf{0.79}_{\pm 0.02}$ | $\mathbf{0.33}_{\pm 0.08}$ | $0.75_{\pm 0.03}$ | $0.32_{\pm 0.11}$ |

Table 1: The $R^2$ score of the conditional cellular responses prediction.

we do not conduct experiments to predict cellular responses to unseen treatments and cell types (covariates). This decision is based on extensive biological research showing that responses to covariates are context-specific (McFarland et al., 2020; Srivatsan et al., 2020). Without thoroughly examining the similarity of unseen treatments or cell types in the latent space, we cannot confidently predict cellular responses.

Nonetheless, our approach enables the prediction of cellular responses at the single-cell level. This paper focuses on predicting cellular responses (expression of 2000 genes) in control cells subjected to drug treatments. It is important to note that our comparative analysis is confined to CPA, VCI, sVAE, scGEN and CINEMA-OT as they are uniquely tailored for this task. We utilize FCR to extract control's $[\mathbf{z}_x^0, \mathbf{z}_{tx}^0, \mathbf{z}_t^0]$ and corresponding experiments' $[\mathbf{z}_x, \mathbf{z}_{tx}, \mathbf{z}_t]$, then use the decoder $g$ to predict the gene expression level as $\hat{\mathbf{y}} = g(\mathbf{z}_x^0, \mathbf{z}_{tx}, \mathbf{z}_t)$. We measure the $R^2$ score. From our results (Table 1), we observe that FCR generally outperforms other baselines across the first three datasets. However, CPA performs the best on the multiPlex-9 dataset. The primary reason for this is that the multiPlex-9 dataset has fewer covariate-specific responses (McFarland et al., 2020). Additionally, scGEN and CINEMA-OT, which are designed for binary perturbations, tend to underperform in these tasks.

## 7 Discussion

This paper aimed to resolve a current challenge—how to disentangle single-cell level drug responses using latent variables into representations for covariates, treatments and contextual covariate-treatment interactions. To do so, we established a theoretically grounded framework for identifiability of such components $\mathbf{z}_{tx}$, $\mathbf{z}_x$ and $\mathbf{z}_t$. Expanding upon these theoretical foundations, we have developed the FCR algorithm to factorize the interactions between treatments and covariates. Looking ahead, our aim is to incorporate interpretable components into this framework. This enhancement will aid in pinpointing genes affected by $\mathbf{z}_x$, $\mathbf{z}_t$ or $\mathbf{z}_{tx}$. Such advancements are expected to significantly contribute to the progress of precision medicine.

## Acknowledgments

The authors would like to extend gratitude to Gregory Barlow for his contribution to the graphic design.

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

# Appendix

# A   Proof of Theorems

## A.1   Proof of Theorem 4.4

**Theorem 4.4**. Let us first define $\mathbf{v}(\mathbf{z}_{tx}, \mathbf{t}, \mathbf{x})$ as the vector:

$$\mathbf{v}(\mathbf{z}_{tx}, \mathbf{t}, \mathbf{x}) = \left[ \frac{\partial q_{n_x+1}(z_{n_x+1}, \mathbf{t}, \mathbf{x})}{\partial z_{n_x+1}}, \cdots, \frac{\partial q_{n_x+n_{tx}}(z_{n_x+n_{tx}}, \mathbf{t}, \mathbf{x})}{\partial z_{n_x+n_{tx}}}, \right.$$

$$\left. \frac{\partial^2 q_{n_x+1}(z_{n_x+1}, \mathbf{t}, \mathbf{x})}{\partial z_{n_x+1}^2}, \cdots, \frac{\partial^2 q_{n_x+n_{tx}}(z_{n_x+n_{tx}}, \mathbf{t}, \mathbf{x})}{\partial z_{n_x+n_{tx}}^2} \right],$$

where $q_i$ denotes the logarithm of probability density $p_{\mathbf{z}_i|\mathbf{t},\mathbf{x}}$ of component $\mathbf{z}_i$. If in addition to the assumptions 4.1, 4.2, 4.3, we assume that for the $2n_{tx}$ vectors

$$\{\mathbf{v}(\mathbf{z}_{tx}, \mathbf{t}_i, \mathbf{x}_i) + \mathbf{v}(\mathbf{z}_{tx}, \mathbf{t}_0, \mathbf{x}_0) - \mathbf{v}(\mathbf{z}_{tx}, \mathbf{t}_0, \mathbf{x}_i) - \mathbf{v}(\mathbf{z}_{tx}, \mathbf{t}_i, \mathbf{x}_0)\}_{i=1}^{2n_{tx}}, \tag{23}$$

are linearly independent, then $\mathbf{z}_{tx}$ is component-wise identifiable.

*Proof.* This proof proceeds in three main steps:

1. **Derivation of the Fundamental System of Equations:** We derive a crucial relationship between the true and estimated latent variables by applying the change of variables formula, and differentiating the equality of observed data distributions.

2. **Isolation of Interactive Components:** We isolate the terms relevant to $\mathbf{z}_{tx}$ by strategically comparing equations for different pairs of $(\mathbf{x}, \mathbf{t})$ values and subtracting them.

3. **Establishing Component-wise Identifiability:** We analyze the structure of the resulting equations and the Jacobian of the transformation between true and estimated latent variables to establish the component-wise identifiability of $\mathbf{z}_{tx}$.

**Step 1 (Derivation of the Fundamental System of Equations)**   Let us assume there exists another latent representation $\hat{\mathbf{z}}$ that yields the same data distribution than the ground-truth variables $\mathbf{z}$, for all $\mathbf{t} \in \mathcal{T}$ and $\mathbf{x} \in \mathcal{X}$. Specifically, we have:

$$p_{\hat{\mathbf{y}}|\mathbf{t},\mathbf{x}} = p_{\mathbf{y}|\mathbf{t},\mathbf{x}}. \tag{24}$$

Given our assumption of noiseless observations, it is equivalent to equality in distribution of the mixed variables:

$$p_{\hat{g}(\hat{\mathbf{z}})|\mathbf{t},\mathbf{x}} = p_{g(\mathbf{z})|\mathbf{t},\mathbf{x}}, \tag{25}$$

which, after a change of variable, is equivalent to:

$$p_{g^{-1}\circ\hat{g}(\hat{\mathbf{z}})|\mathbf{t},\mathbf{x}} \cdot |\mathbf{J}_{g^{-1}}| = p_{\mathbf{z}|\mathbf{t},\mathbf{x}} \cdot |\mathbf{J}_{g^{-1}}|, \tag{26}$$

where $g^{-1} : \mathcal{Y} \to \mathcal{Z}$ denotes the invertible generating function, and $h := g^{-1}\circ\hat{g}$ is the transformation between the true latent variable and estimated one. $|\mathbf{J}_{g^{-1}}|$ denotes the determinant of the Jacobian matrix of $g^{-1}$.

Because $g$ is invertible, $|\mathbf{J}_{g^{-1}}| \neq 0$. Using this fact, we obtain the equivalent condition:

$$p_{h(\hat{\mathbf{z}})|\mathbf{t},\mathbf{x}} = p_{\mathbf{z}|\mathbf{t},\mathbf{x}} \tag{27}$$

According to the independence relations in the data generating process, we have

$$p_{\mathbf{z}|\mathbf{t},\mathbf{x}}(\mathbf{z}|\mathbf{t},\mathbf{x}) = \prod_{i=1}^{n} p_{z_i|\mathbf{t},\mathbf{x}}(z_i \mid \mathbf{t},\mathbf{x}); \qquad p_{\hat{\mathbf{z}}|\mathbf{t},\mathbf{x}}(\hat{\mathbf{z}}|\mathbf{t},\mathbf{x}) = \prod_{i=1}^{n} p_{\hat{z}_i|\mathbf{t},\mathbf{x}}(\hat{z}_i \mid \mathbf{t},\mathbf{x}).$$

Rewriting the notation $q_i := \log p_{z_i|\mathbf{t},\mathbf{x}}$ and $\hat{q}_i := \log p_{\hat{z}_i|\mathbf{t},\mathbf{x}}$ yields:

$$\log p_{\mathbf{z}|\mathbf{t},\mathbf{x}}(\mathbf{z}|\mathbf{t},\mathbf{x}) = \sum_{i=1}^{n} q_i(z_i,\mathbf{t},\mathbf{x}); \qquad \log p_{\hat{\mathbf{z}}|\mathbf{t},\mathbf{x}}(\hat{\mathbf{z}}|\mathbf{t},\mathbf{x}) = \sum_{i=1}^{n} \hat{q}_i(\hat{z}_i,\mathbf{t},\mathbf{x}).$$

Applying the change of variables formula to Equation 24 yields

$$p_{\mathbf{z}|\mathbf{t},\mathbf{x}} = p_{\hat{\mathbf{z}}|\mathbf{t},\mathbf{x}} \cdot |\mathbf{J}_{h^{-1}}| \iff \sum_{i=1}^{n} q_i(z_i, \mathbf{t}, \mathbf{x}) + \log|\mathbf{J}_h| = \sum_{i=1}^{n} \hat{q}_i(\hat{z}_i, \mathbf{t}, \mathbf{x}), \qquad (28)$$

where $\mathbf{J}_{h^{-1}}$ and $\mathbf{J}_h$ are the Jacobian matrix of the transformation associated with $h^{-1}$ and $h$, respectively. We now adopt the following notations,

$$a'_{i,(k)} = \frac{\partial z_i}{\partial \hat{z}_k}, \qquad a''_{i,(k,q)} = \frac{\partial^2 z_i}{\partial \hat{z}_k \partial \hat{z}_q};$$

$$b'_i(z_i, \mathbf{t}, \mathbf{x}) = \frac{\partial q_i(z_i, \mathbf{t}, \mathbf{x})}{\partial z_i}, \qquad b''_i(z_i, \mathbf{t}, \mathbf{x}) = \frac{\partial^2 q_i(z_i, \mathbf{t}, \mathbf{x})}{(\partial z_i)^2}. \qquad (29)$$

Then, we may differentiate Equation 28 with respect to $\hat{z}_k$ and $\hat{z}_q$ where $k, q \in \{1, \ldots, n\}$ and $k \neq q$. Doing so, we obtain the following fundamental system of equations. For any $\mathbf{x} \in \mathcal{X}$, $\mathbf{t} \in \mathcal{T}$, for all $(k, q) \in \{1, \ldots, n\}^2$ such that $k \neq q$:

$$\forall \mathbf{z} \in \mathcal{Z}, \sum_{i=1}^{n} \left[ b''_i(z_i, \mathbf{t}, \mathbf{x}) \cdot a'_{i,(k)} a'_{i,(q)} + b'_i(z_i, \mathbf{t}, \mathbf{x}) a''_{i,(k,q)} \right] + \frac{\partial^2 \log|\mathbf{J}_h|}{\partial \hat{z}_k \hat{z}_q} = 0. \qquad (30)$$

**Step 2 (Isolation of Interactive Components)** We may decompose the sum present on the left-hand-side of Equation 30 across the different block of latent variables, obtaining the following equality:

$$\sum_{i=1}^{n} b''_i(z_i, \mathbf{t}, \mathbf{x}) \cdot a'_{i,(k)} a'_{i,(q)} + b'_i(z_i, \mathbf{t}, \mathbf{x}) a''_{i,(k,q)}$$

$$= \sum_{i=1}^{n_x} b''_i(z_i, \mathbf{x}) \cdot a'_{i,(k)} a'_{i,(q)} + b'_i(z_i, \mathbf{x}) a''_{i,(k,q)}$$

$$+ \sum_{i=n_x+1}^{n_x+n_{tx}} b''_i(z_i, \mathbf{t}, \mathbf{x}) \cdot a'_{i,(k)} a'_{i,(q)} + b'_i(z_i, \mathbf{t}, \mathbf{x}) a''_{i,(k,q)} \qquad (31)$$

$$+ \sum_{i=n_x+n_{tx}+1}^{n} b''_i(z_i, \mathbf{t}) \cdot a'_{i,(k)} a'_{i,(q)} + b'_i(z_i, \mathbf{t}) a''_{i,(k,q)}.$$

Then, we may substitute according to Equation 31 in the fundamental system of equation (30), and strategically apply it to several pairs of environments. We will then take the difference of the systems of equations to make disappear the unknown quantity related to the Jacobian of $h$.

First, we apply this strategy to the pair $\{(\mathbf{x}, \mathbf{t}_0), (\mathbf{x}_0, \mathbf{t}_0)\}$, for any treatment $\mathbf{x}$ (we assume the existence of a reference treatment $\mathbf{t}_0$ and context $\mathbf{x}_0$). Substituting according to Equation 31 into Equation 30, and applying it to $(\mathbf{x}, \mathbf{t}_0)$, we obtain:

$$\sum_{i=1}^{n_x} (b''_i(z_i, \mathbf{x}) \cdot a'_{i,(k)} a'_{i,(q)} + b'_i(z_i, \mathbf{x}) a''_{i,(k,q)}) + \sum_{i=n_x+1}^{n_x+n_{tx}} (b''_i(z_i, \mathbf{t}_0, \mathbf{x}) \cdot a'_{i,(k)} a'_{i,(q)} + b'_i(z_i, \mathbf{t}_0, \mathbf{x}) a''_{i,(k,q)})$$

$$+ \sum_{i=n_x+n_{tx}+1}^{n} (b''_i(z_i, \mathbf{t}_0) \cdot a'_{i,(k)} a'_{i,(q)} + b'_i(z_i, \mathbf{t}_0) a''_{i,(k,q)}) + \frac{\partial^2 \log|\mathbf{J}_h|}{\partial \hat{z}_k \hat{z}_q} = 0. \qquad (32)$$

Proceeding similarly for $(\mathbf{x}_0, \mathbf{t}_0)$, we obtain:

$$\sum_{i=1}^{n_x} (b''_i(z_i, \mathbf{x}_0) \cdot a'_{i,(k)} a'_{i,(q)} + b'_i(z_i, \mathbf{x}_0) a''_{i,(k,q)}) + \sum_{i=n_x+1}^{n_x+n_{tx}} (b''_i(z_i, \mathbf{t}_0, \mathbf{x}_0) \cdot a'_{i,(k)} a'_{i,(q)} + b'_i(z_i, \mathbf{t}_0, \mathbf{x}_0) a''_{i,(k,q)})$$

$$+ \sum_{i=n_x+n_{tx}+1}^{n} (b''_i(z_i, \mathbf{t}_0) \cdot a'_{i,(k)} a'_{i,(q)} + b'_i(z_i, \mathbf{t}_0) a''_{i,(k,q)}) + \frac{\partial^2 \log|\mathbf{J}_h|}{\partial \hat{z}_k \hat{z}_q} = 0. \qquad (33)$$

Then taking the difference between Equation 32 and Equation 33 yields,

$$\sum_{i=1}^{n_x} \left( \left( b_i^{''}(z_i, \mathbf{x}) - b_i^{''}(z_i, \mathbf{x}_0) \right) \cdot a_{i,(k)}^{'} a_{i,(q)}^{'} + \left( b_i^{'}(z_i, \mathbf{x}) - b_i^{'}(z_i, \mathbf{x}_0) \right) a_{i,(k,q)}^{''} \right)$$

$$+ \sum_{i=n_x+1}^{n_x+n_{tx}} \left( \left( b_i^{''}(z_i, \mathbf{t}_0, \mathbf{x}) - b_i^{''}(z_i, \mathbf{t}_0, \mathbf{x}_0) \right) \cdot a_{i,(k)}^{'} a_{i,(q)}^{'} + \left( b_i^{'}(z_i, \mathbf{t}_0, \mathbf{x}) - b_i^{'}(z_i, \mathbf{t}_0, \mathbf{x}_0) \right) a_{i,(k,q)}^{''} \right) = 0. \tag{34}$$

We apply the same principle to the pair $\{(\mathbf{x}_0, \mathbf{t}), (\mathbf{x}_0, \mathbf{t}_0)\}$. We therefore get:

$$\sum_{i=1}^{n_x} (b_i^{''}(z_i, \mathbf{x}_0) \cdot a_{i,(k)}^{'} a_{i,(q)}^{'} + b_i^{'}(z_i, \mathbf{x}_0) a_{i,(k,q)}^{''}) + \sum_{i=n_x+1}^{n_x+n_{tx}} (b_i^{''}(z_i, \mathbf{t}, \mathbf{x}_0) \cdot a_{i,(k)}^{'} a_{i,(q)}^{'} + b_i^{'}(z_i, \mathbf{t}, \mathbf{x}_0) a_{i,(k,q)}^{''})$$

$$+ \sum_{i=n_x+n_{tx}+1}^{n} (b_i^{''}(z_i, \mathbf{t}) \cdot a_{i,(k)}^{'} a_{i,(q)}^{'} + b_i^{'}(z_i, \mathbf{t}) a_{i,(k,q)}^{''}) + \frac{\partial^2 \log |\mathbf{J}_h|}{\partial \hat{z}_k \hat{z}_q} = 0, \tag{35}$$

and,

$$\sum_{i=1}^{n_x} (b_i^{''}(z_i, \mathbf{x}_0) \cdot a_{i,(k)}^{'} a_{i,(q)}^{'} + b_i^{'}(z_i, \mathbf{x}_0) a_{i,(k,q)}^{''}) + \sum_{i=n_x+1}^{n_x+n_{tx}} (b_i^{''}(z_i, \mathbf{t}_0, \mathbf{x}_0) \cdot a_{i,(k)}^{'} a_{i,(q)}^{'} + b_i^{'}(z_i, \mathbf{t}_0, \mathbf{x}_0) a_{i,(k,q)}^{''})$$

$$+ \sum_{i=n_x+n_{tx}+1}^{n} (b_i^{''}(z_i, \mathbf{t}_0) \cdot a_{i,(k)}^{'} a_{i,(q)}^{'} + b_i^{'}(z_i, \mathbf{t}_0) a_{i,(k,q)}^{''}) + \frac{\partial^2 \log |\mathbf{J}_h|}{\partial \hat{z}_k \hat{z}_q} = 0. \tag{36}$$

Taking similarly the difference between Equation 35 and Equation 36 yields:

$$\sum_{i=n_x+1}^{n_x+n_{tx}} \left( \left( b_i^{''}(z_i, \mathbf{t}, \mathbf{x}_0) - b_i^{''}(z_i, \mathbf{t}_0, \mathbf{x}_0) \right) \cdot a_{i,(k)}^{'} a_{i,(q)}^{'} + \left( b_i^{'}(z_i, \mathbf{t}, \mathbf{x}_0) - b_i^{'}(z_i, \mathbf{t}_0, \mathbf{x}_0) \right) a_{i,(k,q)}^{''} \right)$$

$$+ \sum_{i=n_x+n_{tx}+1}^{n} \left( \left( b_i^{''}(z_i, \mathbf{t}) - b_i^{''}(z_i, \mathbf{t}_0) \right) \cdot a_{i,(k)}^{'} a_{i,(q)}^{'} + \left( b_i^{'}(z_i, \mathbf{t}) - b_i^{'}(z_i, \mathbf{t}_0) \right) a_{i,(k,q)}^{''} \right) = 0. \tag{37}$$

Finally, we consider the pairs $\{(\mathbf{x}, \mathbf{t}), (\mathbf{x}_0, \mathbf{t}_0)\}$, for which we obtain the following:

$$\sum_{i=1}^{n_x} (b_i^{''}(z_i, \mathbf{x}) \cdot a_{i,(k)}^{'} a_{i,(q)}^{'} + b_i^{'}(z_i, \mathbf{x}) a_{i,(k,q)}^{''}) + \sum_{i=n_x+1}^{n_x+n_{tx}} (b_i^{''}(z_i, \mathbf{t}, \mathbf{x}) \cdot a_{i,(k)}^{'} a_{i,(q)}^{'} + b_i^{'}(z_i, \mathbf{t}, \mathbf{x}) a_{i,(k,q)}^{''})$$

$$+ \sum_{i=n_x+n_{tx}+1}^{n} (b_i^{''}(z_i, \mathbf{t}) \cdot a_{i,(k)}^{'} a_{i,(q)}^{'} + b_i^{'}(z_i, \mathbf{t}) a_{i,(k,q)}^{''}) + \frac{\partial^2 \log |\mathbf{J}_h|}{\partial \hat{z}_k \hat{z}_q} = 0, \tag{38}$$

and

$$\sum_{i=1}^{n_x} (b_i^{''}(z_i, \mathbf{x}_0) \cdot a_{i,(k)}^{'} a_{i,(q)}^{'} + b_i^{'}(z_i, \mathbf{x}_0) a_{i,(k,q)}^{''}) + \sum_{i=n_x+1}^{n_x+n_{tx}} (b_i^{''}(z_i, \mathbf{t}_0, \mathbf{x}_0) \cdot a_{i,(k)}^{'} a_{i,(q)}^{'} + b_i^{'}(z_i, \mathbf{t}_0, \mathbf{x}_0) a_{i,(k,q)}^{''})$$

$$+ \sum_{i=n_x+n_{tx}+1}^{n} (b_i^{''}(z_i, \mathbf{t}_0) \cdot a_{i,(k)}^{'} a_{i,(q)}^{'} + b_i^{'}(z_i, \mathbf{t}_0) a_{i,(k,q)}^{''}) + \frac{\partial^2 \log |\mathbf{J}_h|}{\partial \hat{z}_k \hat{z}_q} = 0. \tag{39}$$

Once again taking the difference between Equation 38 and Equation 39 yields:

$$\sum_{i=1}^{n_x} \left( \left( b_i^{''}(z_i, \mathbf{x}) - b_i^{''}(z_i, \mathbf{x}_0) \right) \cdot a_{i,(k)}^{'} a_{i,(q)}^{'} + \left( b_i^{'}(z_i, \mathbf{x}) - b_i^{'}(z_i, \mathbf{x}_0) \right) a_{i,(k,q)}^{''} \right)$$

$$+ \sum_{i=n_x+1}^{n_x+n_{tx}} \left( (b_i^{''}(z_i, \mathbf{t}, \mathbf{x}) - b_i^{''}(z_i, \mathbf{t}_0, \mathbf{x}_0)) \cdot a_{i,(k)}^{'} a_{i,(q)}^{'} + \left( b_i^{'}(z_i, \mathbf{t}, \mathbf{x}) - b_i^{'}(z_i, \mathbf{t}_0, \mathbf{x}_0) \right) a_{i,(k,q)}^{''} \right)$$

$$+ \sum_{i=n_x+n_{tx}+1}^{n} \left( (b_i^{''}(z_i, \mathbf{t}) - b_i^{''}(z_i, \mathbf{t}_0)) \cdot a_{i,(k)}^{'} a_{i,(q)}^{'} + \left( b_i^{'}(z_i, \mathbf{t}, \mathbf{x}) - b_i^{'}(z_i, \mathbf{t}_0) \right) a_{i,(k,q)}^{''} \right) = 0. \tag{40}$$

As a final step, we are going to combine Equations 34, 37 and 40 in order to correctly isolate the interaction components.

We take the difference between Equation 40 and Equation 34:

$$\sum_{i=n_x+1}^{n_x+n_{tx}} \left[ \left( (b_i^{''}(z_i, \mathbf{t}, \mathbf{x}) - b_i^{''}(z_i, \mathbf{t}_0, \mathbf{x}_0)) \right) - \left( (b_i^{''}(z_i, \mathbf{t}_0, \mathbf{x}) - b_i^{''}(z_i, \mathbf{t}_0, \mathbf{x}_0)) \right) \right] \cdot a_{i,(k)}^{'} a_{i,(q)}^{'}$$

$$+ \left[ \left( b_i^{'}(z_i, \mathbf{t}, \mathbf{x}) - b_i^{'}(z_i, \mathbf{t}_0, \mathbf{x}_0) \right) - \left( b_i^{'}(z_i, \mathbf{t}_0, \mathbf{x}) - b_i^{'}(z_i, \mathbf{t}_0, \mathbf{x}_0) \right) \right] a_{i,(k,q)}^{''})$$

$$+ \sum_{i=n_x+n_{tx}+1}^{n} \left( \left( b_i^{''}(z_i, \mathbf{t}) - b_i^{''}(z_i, , \mathbf{t}_0) \right) \cdot a_{i,(k)}^{'} a_{i,(q)}^{'} + \left( b_i^{'}(z_i, \mathbf{t}) - b_i^{'}(z_i, \mathbf{t}_0) \right) a_{i,(k,q)}^{''} \right) = 0 \tag{41}$$

Finally, we subtract Equation 41 and Equation 37:

$$\sum_{i=n_x+1}^{n_x+n_{tx}} \left[ \left( (b_i^{''}(z_i, \mathbf{t}, \mathbf{x}) - b_i^{''}(z_i, \mathbf{t}_0, \mathbf{x}_0)) \right) - \left( (b_i^{''}(z_i, \mathbf{t}_0, \mathbf{x}) - b_i^{''}(z_i, \mathbf{t}_0, \mathbf{x}_0)) \right) - \left( b_i^{''}(z_i, \mathbf{t}, \mathbf{x}_0) - b_i^{''}(z_i, \mathbf{t}_0, \mathbf{x}_0) \right) \right] \cdot a_{i,(k)}^{'} a_{i,(q)}^{'}$$

$$+ \left[ \left( b_i^{'}(z_i, \mathbf{t}, \mathbf{x}) - b_i^{'}(z_i, \mathbf{t}_0, \mathbf{x}_0) \right) - \left( b_i^{'}(z_i, \mathbf{t}_0, \mathbf{x}) - b_i^{'}(z_i, \mathbf{t}_0, \mathbf{x}_0) \right) - \left( b_i^{'}(z_i, \mathbf{t}, \mathbf{x}_0) - b_i^{'}(z_i, \mathbf{t}_0, \mathbf{x}_0) \right) \right] a_{i,(k,q)}^{''} = 0.$$

This last equation may be rearranged as:

$$\sum_{i=n_x+1}^{n_x+n_{tx}} \left[ b_i^{''}(z_i, \mathbf{t}, \mathbf{x}) - b_i^{''}(z_i, \mathbf{t}_0, \mathbf{x}) - b_i^{''}(z_i, \mathbf{t}, \mathbf{x}_0) + b_i^{''}(z_i, \mathbf{t}_0, \mathbf{x}_0) \right] \cdot a_{i,(k)}^{'} a_{i,(q)}^{'}$$

$$+ \left[ b_i^{'}(z_i, \mathbf{t}, \mathbf{x}) - b_i^{'}(z_i, \mathbf{t}_0, \mathbf{x}) - b_i^{'}(z_i, \mathbf{t}, \mathbf{x}_0) + b_i^{'}(z_i, \mathbf{t}_0, \mathbf{x}_0) \right] a_{i,(k,q)}^{''} = 0. \tag{42}$$

**Step 3 (Establishing Component-wise Identifiability)** Given the assumption of linear independence in the Theorem, the linear system is a $2n_{tx} \times 2n_{tx}$ full-rank system. Therefore, the only solution is:

$$\begin{cases} a_{i,(k)}^{'} a_{i,(q)}^{'} = 0 \\ a_{i,(k,q)}^{''} = 0 \end{cases} \quad \text{for } i \in \{n_x+1, \ldots, n_x+n_{tx}\} \text{ and } k, q \in \{1, \ldots, n\}, k \neq q.$$

$h(\cdot)$ is a smooth function over $\mathcal{Z}$ and its Jacobian can written as:

$$\mathbf{J}_h = \begin{bmatrix} \mathbf{A} := \frac{\partial \mathbf{z}_x}{\partial \hat{\mathbf{z}}_x} & \mathbf{B} := \frac{\partial \mathbf{z}_x}{\partial \mathbf{z}_{tx}^{'}} & \mathbf{C} := \frac{\partial \mathbf{z}_x}{\partial \hat{\mathbf{z}}_t} \\ \mathbf{D} := \frac{\partial \mathbf{z}_{tx}}{\partial \hat{\mathbf{z}}_x} & \mathbf{E} := \frac{\partial \mathbf{z}_{tx}}{\partial \mathbf{z}_{tx}^{'}} & \mathbf{F} := \frac{\partial \mathbf{z}_{tx}}{\partial \hat{\mathbf{z}}_t} \\ \mathbf{G} := \frac{\partial \mathbf{z}_t}{\partial \hat{\mathbf{z}}_x} & \mathbf{H} := \frac{\partial \mathbf{z}_t}{\partial \mathbf{z}_{tx}^{'}} & \mathbf{I} := \frac{\partial \mathbf{z}_t}{\partial \hat{\mathbf{z}}_t} \end{bmatrix}. \tag{43}$$

Note that $a_{i,(k)}^{'} a_{i,(q)}^{'} = 0$ implies that for each $i \in \{n_x+1, \ldots, n_x+n_{tx}\}$, $a_{i,(k)}^{'} \neq 0$ for at most one element $k \in [n]$. As a consequence, there's only a single non-zero entry in each row indexed by $i \in \{n_x+1, \ldots, n_x+n_{tx}\}$ in the Jacobian matrix $\mathbf{J}_h$. Further strengthening this argument, the invertibility of $h$ requires $\mathbf{J}_h$ to be full-rank, suggesting there's precisely one non-zero component in each row of matrices $\mathbf{D}$, $\mathbf{E}$, and $\mathbf{F}$.

It means that each of $z_i \in \mathbf{z}_{tx}$ for $i \in \{n_x+1, \ldots, n_x+n_{tx}\}$ is attributed to at most one of the $\hat{\mathbf{z}}$. And because the other $\hat{z}$ that are not in the block $\hat{\mathbf{z}}_{tx}$ do not have dependencies with both $t$ and $x$, it must be that the non-zero coefficient is in the block $\mathbf{E}$. Therefore $\mathbf{D} = \mathbf{0}$ and $\mathbf{F} = \mathbf{0}$. This indicates the invertibility of $h_i$ for every $i$ in the range $\{n_x+1, \ldots, n_x+n_{tx}\}$. In conclusion, $\mathbf{z}_{tx}$ are element-wise identifiable, albeit subject to permutations and component-wise invertible transformations. $\qquad\square$

## A.2 Proof of Theorem 4.5

**Theorem 4.5:** We follow Assumptions 4.1, 4.2, 4.3, and the one from Theorem 4.4. We note as $\mathcal{S}(Z)$ the set of subsets $S \subseteq \mathcal{Z}$ of $\mathcal{Z}$ that satisfy the following two conditions:

   (i) $S$ has nonzero probability measure, i.e. $\mathbb{P}(\mathbf{z} \in S \mid \mathbf{t} = \mathbf{t}', \mathbf{x} = \mathbf{x}') > 0$ for any $\mathbf{t}' \in \mathcal{T}$ and $\mathbf{x}' \in \mathcal{X}$.

   (ii) $S$ cannot be expressed as $A_{\mathbf{z}_x} \times \mathcal{Z}_{tx} \times \mathcal{Z}_t$ for any $A_{\mathbf{z}_x} \subset \mathcal{Z}_x$ or as $\mathcal{Z}_x \times \mathcal{Z}_{tx} \times A_{\mathbf{z}_t}$ for any $A_{\mathbf{z}_t} \subset \mathcal{Z}_t$.

We have the following identifiability result. If for all $S \in \mathcal{S}(\mathcal{Z})$, there exists $(\mathbf{t}_1, \mathbf{t}_2) \in \mathcal{T} \times \mathcal{T}$ and $\mathbf{x} \in \mathcal{X}$ such that

$$\int_{\mathbf{z} \in S} p_{\mathbf{z}|\mathbf{t},\mathbf{x}}(\mathbf{z} \mid \mathbf{t}_1, \mathbf{x})d\mathbf{z} \neq \int_{\mathbf{z} \in S} p_{\mathbf{z}|\mathbf{t},\mathbf{x}}(\mathbf{z} \mid \mathbf{t}_2, \mathbf{x})d\mathbf{z}, \tag{44}$$

and there also exists $(\mathbf{x}_1, \mathbf{x}_2) \in \mathcal{X} \times \mathcal{X}$ and $\mathbf{t} \in \mathcal{T}$ such that

$$\int_{\mathbf{z} \in S} p_{\mathbf{z}|\mathbf{t},\mathbf{x}}(\mathbf{z} \mid \mathbf{t}, \mathbf{x}_1)d\mathbf{z} \neq \int_{\mathbf{z} \in S} p_{\mathbf{z}|\mathbf{t},\mathbf{x}}(\mathbf{z} \mid \mathbf{t}, \mathbf{x}_2)d\mathbf{z}, \tag{45}$$

then $\mathbf{z}_t$ and $\mathbf{z}_x$ are block-wise identifiable.

*Proof.* The proof of the block-wise identifiability of $\mathbf{z}_x$ and $\mathbf{z}_t$ are similar, so here we focus the proof about the identifiability of $\mathbf{z}_x$. Our proof draws parallels to the approach in Kong et al. (2022), but our context specifically pertains to the multi-conditions distributions scenarios. Throughout, we use the notations $\mathbf{z}_t^+ = [\mathbf{z}_{tx}, \mathbf{z}_t]$, $\hat{\mathbf{z}}_t^+ = [\hat{\mathbf{z}}_{tx}, \hat{\mathbf{z}}_t]$ for simplification. This proof is comprised of four major steps.

   1. **Integral characterization of domain invariance**. We first leverage properties of the generating process and the marginal distribution matching condition to provide a characterization of the invariance of a block of latent variables by a mixing function using an integral condition.

   2. **Topological characterization of invariance**. We derive equivalence statements for domain invariance of functions.

   3. **Proof of invariance by contradiction**. We prove the invariance statement from Step 2 by contradiction. Specifically, we show that if $\hat{\mathbf{z}}_x$ depended on $\mathbf{z}_t^+$, the invariance derived in Step 1 would break.

   4. **Block-wise identifiability of $\mathbf{z}_t$ and $\mathbf{z}_x$**. We use the conclusion in Step 3, the regularity properties of $h$, and the conclusion in Theorem 4.4 to show the identifiability result.

**Step 1 (Integral characterization of domain invariance).** As a reminder to the reader, $g : \mathcal{Z} \to \mathcal{Y}$ denotes the ground-truth mixing function, and $\hat{g} : \mathcal{Z} \to \mathcal{Y}$ denotes the learned mixing function. We assume that both $g$ and $\hat{g}$ are invertible, so that their reciprocal functions are well defined. In particular, we denote by $\hat{g}_{1:n_x}^{-1} : \mathcal{Y} \to \mathcal{Z}_x$ the estimated transformation from the observation to the covariate-specific block of the latent space $\mathcal{Z}$. We seek to find an integral characterization of the invariance of the learned function $\hat{g}$ on the domain $\mathcal{Z}_x$.

We seek to derive a condition for which the distribution of latent variables on $\mathcal{Z}_x$ will be unchanged when the treatment $\mathbf{t}$ is changed (it will however depend on $\mathbf{x}$). Let $S \subset \mathcal{Z}_x$ designates a subset of $\mathcal{Z}_x$ and $\mathbf{x} \in \mathcal{X}$ be any context. We seek to characterize the following condition:

$$\forall(\mathbf{t}_1, \mathbf{t}_2) \in \mathcal{T}^2, \mathbb{P}\left[\{\hat{g}_{1:n_x}^{-1}(\hat{\mathbf{y}}) \in S\} \mid \{\mathbf{x}, \mathbf{t} = \mathbf{t}_1\}\right] = \mathbb{P}\left[\{\hat{g}_{1:n_x}^{-1}(\hat{\mathbf{y}}) \in S\} \mid \{\mathbf{x}, \mathbf{t} = \mathbf{t}_2\}\right]. \tag{46}$$

This condition can be written equivalently using the pre-image set of $S$:

$$\forall (\mathbf{t}_1, \mathbf{t}_2) \in \mathcal{T}^2, \mathbb{P}\left[\left\{\hat{\mathbf{y}} \in \left(\hat{g}_{1:n_x}^{-1}\right)^{-1}(S)\right\} \mid \{\mathbf{x}, \mathbf{t} = \mathbf{t}_1\}\right] = \mathbb{P}\left[\left\{\hat{\mathbf{y}} \in \left(\hat{g}_{1:n_x}^{-1}\right)^{-1}(S)\right\} \mid \{\mathbf{x}, \mathbf{t} = \mathbf{t}_2\}\right], \tag{47}$$

where $\left(\hat{g}_{1:n_x}^{-1}\right)^{-1}(S) \subseteq \mathcal{Y}$ is the set of estimated observations $\hat{\mathbf{y}}$ originating from covariate-specific variables $\hat{\mathbf{z}}_x$ in $S$.

Because of the equality of the observed data $\mathbf{y}$ and the generated data distribution from the estimated model $\hat{\mathbf{y}}$, the relation in Equation 47 also holds for the random variable $\mathbf{y}$

$$\forall (\mathbf{t}_1, \mathbf{t}_2) \in \mathcal{T}^2, \mathbb{P}\left[\left\{\mathbf{y} \in \left(\hat{g}_{1:n_x}^{-1}\right)^{-1}(S)\right\} \mid \{\mathbf{x}, \mathbf{t} = \mathbf{t}_1\}\right] = \mathbb{P}\left[\left\{\mathbf{y} \in \left(\hat{g}_{1:n_x}^{-1}\right)^{-1}(S)\right\} \mid \{\mathbf{x}, \mathbf{t} = \mathbf{t}_2\}\right]. \tag{48}$$

It follows, by applying the image of the function $\hat{g}_{1:n_x}^{-1}$, that:

$$\forall (\mathbf{t}_1, \mathbf{t}_2) \in \mathcal{T}^2, \mathbb{P}\left[\left\{\hat{g}_{1:n_x}^{-1}(\mathbf{y}) \in S\right\} \mid \{\mathbf{x}, \mathbf{t} = \mathbf{t}_1\}\right] = \mathbb{P}\left[\left\{\hat{g}_{1:n_x}^{-1}(\mathbf{y}) \in S\right\} \mid \{\mathbf{x}, \mathbf{t} = \mathbf{t}_2\}\right]. \tag{49}$$

Since $g$ and $\hat{g}$ are smooth and injective, we may define the function $\bar{h} = \hat{g}^{-1} \circ g : \mathcal{Z} \to \mathcal{Z}$. We note that by definition $\bar{h} = h^{-1}$ where $h$ is introduced in the proof of Theorem 4.4. We now remind the reader that $\mathbf{y} = g(\mathbf{z})$. Therefore, using the notation $\bar{h}_x := \bar{h}_{1:n_x} : \mathcal{Z} \to \mathcal{Z}_x$, we have the equivalent condition

$$\forall (\mathbf{t}_1, \mathbf{t}_2) \in \mathcal{T}^2, \mathbb{P}\left[\left\{\bar{h}_x(\mathbf{z}) \in S\right\} \mid \{\mathbf{x}, \mathbf{t} = \mathbf{t}_1\}\right] = \mathbb{P}\left[\left\{\bar{h}_x(\mathbf{z}) \in S\right\} \mid \{\mathbf{x}, \mathbf{t} = \mathbf{t}_2\}\right]. \tag{50}$$

Now, using the pre-image formulation again, we may write it as

$$\forall (\mathbf{t}_1, \mathbf{t}_2) \in \mathcal{T}^2, \mathbb{P}\left[\left\{\mathbf{z} \in \bar{h}_x^{-1}(S)\right\} \mid \{\mathbf{x}, \mathbf{t} = \mathbf{t}_1\}\right] = \mathbb{P}\left[\left\{\mathbf{z} \in \bar{h}_x^{-1}(S)\right\} \mid \{\mathbf{x}, \mathbf{t} = \mathbf{t}_2\}\right], \tag{51}$$

and, using an integral notation:

$$\forall (\mathbf{t}_1, \mathbf{t}_2) \in \mathcal{T}^2, \int_{\mathbf{z} \in \bar{h}_x^{-1}(S)} p_{\mathbf{z}|\mathbf{t}, \mathbf{x}}(\mathbf{z} \mid \mathbf{x}, \mathbf{t}_1) \, d\mathbf{z} = \int_{\mathbf{z} \in \bar{h}_x^{-1}(S)} p_{\mathbf{z}|\mathbf{t}, \mathbf{x}}(\mathbf{z} \mid \mathbf{x}, \mathbf{t}_2) \, d\mathbf{z}, \tag{52}$$

where $\bar{h}_x^{-1}(S) = \{\mathbf{z} \in \mathcal{Z} : \bar{h}_x(\mathbf{z}) \in S\}$ is the pre-image of $S$.

By exploiting the factorization of the likelihood, we obtain our final condition, equivalent to the one in Equation 47 for any $S \subseteq \mathcal{Z}_x$ and $\mathbf{x} \in \mathcal{X}$:

$$\int_{[\mathbf{z}_x, \mathbf{z}_t^+] \in \bar{h}_x^{-1}(S)} p_{\mathbf{z}_x|\mathbf{x}}(\mathbf{z}_\mathbf{x} \mid \mathbf{x}) \left( p_{\mathbf{z}_t^+|\mathbf{x}, \mathbf{t}}\left(\mathbf{z}_t^+ \mid \mathbf{x}, \mathbf{t}_1\right) - p_{\mathbf{z}_t^+|\mathbf{x}, \mathbf{t}}\left(\mathbf{z}_t^+ \mid \mathbf{x}, \mathbf{t}_2\right) \right) d\mathbf{z}_x d\mathbf{z}_t^+ = 0, \tag{53}$$

where the condition must hold for all $(\mathbf{t}_1, \mathbf{t}_2) \in \mathcal{T}^2$.

**Step 2 (Topological characterization of invariance).** To demonstrate the block-identifiability of $\mathbf{z}_x$, our objective is to substantiate that $\bar{h}_x([\mathbf{z}_x, \mathbf{z}_{tx}, \mathbf{z}_t]) = \bar{h}_x([\mathbf{z}_x, \mathbf{z}_t^+])$ is functionally independent of $\mathbf{z}_t^+$. To achieve this, we initially formulate a set of equivalent statement:

1. **Statement 1**. $\bar{h}_x\left(\left[\mathbf{z}_x^\top, \mathbf{z}_t^{+\top}\right]^\top\right)$ does not depend on $\mathbf{z}_t^+$.

2. **Statement 2**. $\forall \mathbf{z}_x \in \mathcal{Z}_x, \exists B_{\mathbf{z}_x} \subseteq \mathcal{Z}_x \setminus \emptyset : \bar{h}_x^{-1}(\mathbf{z}_x) = B_{\mathbf{z}_x} \times \mathcal{Z}_{tx} \times \mathcal{Z}_t$.

3. **Statement 3**. $\forall \mathbf{z}_x \in \mathcal{Z}_x, \forall r \in \mathbb{R}^+, \exists B_{\mathbf{z}_x}^+ \subseteq \mathcal{Z}_x \setminus \emptyset : \bar{h}_x^{-1}(\mathcal{B}_r(\mathbf{z}_x)) = B_{\mathbf{z}_x}^+ \times \mathcal{Z}_{tx} \times \mathcal{Z}_t$,

where $\mathcal{B}_r(\mathbf{z}_x)$ is defined as the ball centered around $\mathbf{z}_x$ with radius $r$: $\mathcal{B}_r(\mathbf{z}_x) = \left\{\mathbf{z}_x' \in \mathcal{Z}_x : \|\mathbf{z}_x' - \mathbf{z}_x\|^2 < r\right\}$.

We note that Statement 2 is a mathematical formulation of Statement 1, and that Statement 3 is a generalization of Statement 2 from singletons $\{\mathbf{z}_x\}$ in Statement 2 to open, non-empty balls $\mathcal{B}_r(\mathbf{z}_x)$. We proceed to demonstrating equivalence between those statements.

**Statement 2 $\Rightarrow$ Statement 3.** Let $\mathbf{z}_x \in \mathcal{Z}_x$ and $r \in \mathbb{R}^+$. By definition of the pre-image of a set, we have that:

$$\bar{h}_x^{-1}(\mathcal{B}_r(\mathbf{z}_x)) = \cup_{\mathbf{z}_x' \in \mathcal{B}_r(\mathbf{z}_x)} h_x^{-1}(\mathbf{z}_x'). \tag{54}$$

Because we assume Statement 2, we have that for all $\mathbf{z}_x' \in \mathcal{B}_r(\mathbf{z}_x)$, there exists a set $B_{\mathbf{z}_x'}$ such that $h_x^{-1}(\mathbf{z}_x') = B_{\mathbf{z}_x'} \times \mathcal{Z}_{tx} \times \mathcal{Z}_t$. Therefore, Statement 3 stands for $B_{\mathbf{z}_x}^+ = \cup_{\mathbf{z}_x'} B_{\mathbf{z}_x'}$.

**Statement 2 ⇐ Statement 3.** We proceed by contradiction. Suppose that Statement 2 is false, then for a certain $\bar{\mathbf{z}}_x^* \in \mathcal{Z}_x$, it is possible to construct a point $\bar{\mathbf{z}}^B = [\bar{\mathbf{z}}_x^B, \bar{\mathbf{z}}_{tx}^B, \bar{\mathbf{z}}_t^B] \in \mathcal{Z}$ such that $\bar{\mathbf{z}}_x^B$ is in the pre-image of $\{\bar{\mathbf{z}}_x^*\}$ by $\bar{h}_x^{-1}$ but $\bar{h}_x\left(\bar{\mathbf{z}}^B\right) \neq \bar{\mathbf{z}}_x$. Indeed, this directly means that changing the other components of $\mathcal{Z}$ at the input of $\bar{h}$ can alter the component of $\mathcal{Z}_x$ at its output. By continuity of $\bar{h}_x$, there exists $\hat{r} > 0$ such that $\bar{h}_x\left(\bar{\mathbf{z}}^B\right) \notin \mathcal{B}_{\hat{r}}\left(\bar{\mathbf{z}}_x\right)$. For such $\hat{r}$, we have that $\bar{\mathbf{z}}^B \notin \bar{h}_x^{-1}\left(\mathcal{B}_{\hat{r}}\left(\bar{\mathbf{z}}_x\right)\right)$. Additionally, the application of Statement 3 suggests that there exists a non-trivial subset $B_{\mathbf{z}_x}^+$ such that $h_x^{-1}\left(\mathcal{B}_{\hat{r}}\left(\bar{\mathbf{z}}_x\right)\right) = B_{\mathbf{z}_x}^+ \times \mathcal{Z}_{tx} \times \mathcal{Z}_t$. By definition of $\bar{\mathbf{z}}^B$, it is clear that $\bar{\mathbf{z}}_{1:n_x}^B \in B_{\mathbf{z}_x}^+$. The fact that $\bar{\mathbf{z}}^B \notin h_x^{-1}\left(\mathcal{B}_{\hat{r}}\left(\bar{\mathbf{z}}_x\right)\right)$ contradicts Statement 3. Therefore, Statement 2 is true under the premise of Statement 3.

**Step 3 (Proof of invariance by contradiction).** We first show that the pre-image of any open balls of $\mathcal{Z}_x$ are non-empty and open sets. For $\mathbf{z}_x \in \mathcal{Z}_x$ and $r \in \mathbb{R}^+$, we note that because $\mathcal{B}_r\left(\mathbf{z}_x\right)$ is open and $\bar{h}_x$ is continuous, the pre-image $\bar{h}_x^{-1}\left(\mathcal{B}_r\left(\mathbf{z}_x\right)\right)$ is open. In addition, because $h$ is continuous and we have equality of the generated data distributions:

$$\forall \mathbf{t} \in \mathcal{T}, \forall \mathbf{x} \in \mathcal{X}, \mathbb{P}\left[\{\mathbf{y} \in S\} \mid \{\mathbf{t}, \mathbf{x}\}\right] = \mathbb{P}\left[\{\hat{\mathbf{y}} \in S\} \mid \{\mathbf{t}, \mathbf{x}\}\right], \tag{55}$$

we have that $h$ is a bijection (Klindt et al., 2021), which ensures that $\bar{h}_x^{-1}\left(\mathcal{B}_r\left(\mathbf{z}_x\right)\right)$ is non-empty. Hence, $\bar{h}_x^{-1}\left(\mathcal{B}_r\left(\mathbf{z}_x\right)\right)$ is both non-empty and open.

We now assume, by contradiction, that $\mathbf{z}_x$ is not block-identifiable. Therefore, $\bar{h}_x$ is not invariant with respect to $\mathbf{z}_t^+$. Additionally, because of Step 2, we have the existence of a ball $S^* := \mathcal{B}_{r^*}\left(\mathbf{z}_x^*\right)$ centered on the point $\mathbf{z}_x^* \in \mathcal{Z}_x$ and of radius $r^* \in \mathbb{R}^+$ such that $\bar{h}_x^{-1}\left(S^*\right)$ cannot be written of the form $A \times \mathcal{Z}_{tx} \times \mathcal{Z}_t$ for any non-trivial $A \subset \mathcal{Z}_x$.

We may therefore define the set $B_{\mathbf{z}}^* := \left\{\mathbf{z} \in \bar{h}_x^{-1}\left(S^*\right) \mid \{\mathbf{z}_{1:n_x}\} \times \mathcal{Z}_t \times \mathcal{Z}_{tx} \not\subseteq \bar{h}_x^{-1}\left(S^*\right)\right\}$. Intuitively, $B_{\mathbf{z}}^*$ contains the partition of the pre-image $\bar{h}_x^{-1}\left(S^*\right)$ that the $\mathbf{t}$ part $\mathbf{z}_t^+$ cannot take on any value in $\mathcal{Z}_{tx} \times \mathcal{Z}_t$. It is therefore non-empty by hypothesis. To show contradiction with Equation 53, we evaluate it on the set $S^*$ and split the integral on two domains of the partition $\bar{h}_x^{-1}\left(S^*\right) = \left(\bar{h}_x^{-1}\left(S^*\right) \setminus B_{\mathbf{z}}^*\right) \cup B_{\mathbf{z}}^*$.

We define the following integrals:

$$T = \int_{\left[\mathbf{z}_x, \mathbf{z}_t^+\right] \in \bar{h}_x^{-1}(S^*)} p_{\mathbf{z}_x \mid \mathbf{x}}\left(\mathbf{z}_\mathbf{x} \mid \mathbf{x}\right) \left(p_{\mathbf{z}_t^+ \mid \mathbf{x}, \mathbf{t}}\left(\mathbf{z}_t^+ \mid \mathbf{x}, \mathbf{t}_1\right) - p_{\mathbf{z}_t^+ \mid \mathbf{x}, \mathbf{t}}\left(\mathbf{z}_\mathbf{t}^+ \mid \mathbf{x}, \mathbf{t}_2\right)\right) d\mathbf{z}_x d\mathbf{z}_t^+ \tag{56}$$

$$T_1 = \int_{\left[\mathbf{z}_x, \mathbf{z}_t^+\right] \in \bar{h}_x^{-1}(S) \setminus B_{\mathbf{z}}^*} p_{\mathbf{z}_x \mid \mathbf{x}}\left(\mathbf{z}_\mathbf{x} \mid \mathbf{x}\right) \left(p_{\mathbf{z}_t^+ \mid \mathbf{t}, \mathbf{x}}\left(\mathbf{z}_t^+ \mid \mathbf{x}, \mathbf{t}_1\right) - p_{\mathbf{z}_t^+ \mid \mathbf{t}, \mathbf{x}}\left(\mathbf{z}_t^+ \mid \mathbf{x}, \mathbf{t}_2\right)\right) d\mathbf{z}_x d\mathbf{z}_t^+$$
$$\tag{57}$$

$$T_2 = \int_{\left[\mathbf{z}_x, \mathbf{z}_t^+\right] \in B_{\mathbf{z}}^*} p_{\mathbf{z}_x \mid \mathbf{x}}\left(\mathbf{z}_\mathbf{x} \mid \mathbf{x}\right) \left(p_{\mathbf{z}_t^+ \mid \mathbf{t}, \mathbf{x}}\left(\mathbf{z}_t^+ \mid \mathbf{x}, \mathbf{t}_1\right) - p_{\mathbf{z}_t^+ \mid \mathbf{t}, \mathbf{x}}\left(\mathbf{z}_t^+ \mid \mathbf{x}, \mathbf{t}_2\right)\right) d\mathbf{z}_x d\mathbf{z}_t^+, \tag{58}$$

where we have the expected relation $T = T_1 + T_2$.

We first look at the value of $T_1$. In the case where the set $\bar{h}_x^{-1}\left(S^*\right) \setminus B_{\mathbf{z}}^*$ is empty, then $T_1$ trivially evaluates to 0. Otherwise, there exists a non-empty subset $C_{\mathbf{z}_x}^*$ of $\mathcal{Z}_x$ such that $\bar{h}_x^{-1}\left(S^*\right) \setminus B_{\mathbf{z}}^* = C_{\mathbf{z}_x}^* \times \mathcal{Z}_{tx} \times \mathcal{Z}_t$. With this expression, it follows that

$$T_1 = \int_{\left[\mathbf{z}_x, \mathbf{z}_t^+\right] \in C_{\mathbf{z}_x}^* \times \mathcal{Z}_{tx} \times \mathcal{Z}_t} p_{\mathbf{z}_x \mid \mathbf{x}}\left(\mathbf{z}_\mathbf{x} \mid \mathbf{x}\right) \left(p_{\mathbf{z}_t^+ \mid \mathbf{x}, \mathbf{t}}\left(\mathbf{z}_t^+ \mid \mathbf{x}, \mathbf{t}_1\right) - p_{\mathbf{z}_t^+ \mid \mathbf{x}, \mathbf{t}}\left(\mathbf{z}_t^+ \mid \mathbf{x}, \mathbf{t}_2\right)\right) d\mathbf{z}_x d\mathbf{z}_t^+.$$
$$\tag{59}$$

Because of the separability of the domains, we may apply Fubini's theorem:

$$T_1 = \int_{\mathbf{z}_x \in C_{\mathbf{z}_x}^*} p_{\mathbf{z}_x \mid \mathbf{x}}\left(\mathbf{z}_x \mid \mathbf{x}\right) d\mathbf{z}_x \int_{\mathbf{z}_t^+ \in \mathcal{Z}_{tx} \times \mathcal{Z}_t} \left(p_{\mathbf{z}_t^+ \mid \mathbf{x}, \mathbf{t}}\left(\mathbf{z}_t^+ \mid \mathbf{x}, \mathbf{t}_1\right) - p_{\mathbf{z}_t^+ \mid \mathbf{x}, \mathbf{t}}\left(\mathbf{z}_t^+ \mid \mathbf{x}, \mathbf{t}_2\right)\right) d\mathbf{z}_x d\mathbf{z}_t^+$$
$$\tag{60}$$

$$T_1 = \int_{\mathbf{z}_x \in C_{\mathbf{z}_x}^*} p_{\mathbf{z}_x \mid \mathbf{x}}\left(\mathbf{z}_{x \mid \mathbf{x}}\right) (1 - 1) d\mathbf{z}_x = 0. \tag{61}$$

Therefore, in both cases $T_1$ evaluates to 0 for $S^*$.

Now, we address $T_2$. Towards this goal, we prove that $B_{\mathbf{z}}^*$ satisfies the condition for application of the assumption of the theorem. First, we must show that $B_{\mathbf{z}}^*$ has non-zero probability measure for all values of $\mathbf{t}$ and $\mathbf{x}$. For this, it is enough to show that $B_{\mathbf{z}}^*$ contains an open set, given that we assume that $p_{\mathbf{z}|\mathbf{t},\mathbf{x}}(\mathbf{z} \mid \mathbf{t}, \mathbf{x}) > 0$ over $(\mathbf{z}, \mathbf{t}, \mathbf{x}) \in \mathcal{Z} \times \mathcal{T} \times \mathcal{X}$. Let us take one element $\mathbf{z}_B \in B_{\mathbf{z}}^*$, which is possible because we proved $B_{\mathbf{z}}^*$ is non-empty. As discussed above, $\bar{h}_x^{-1}(S^*)$ is open and non-empty, and by continuity of $\bar{h}_x$, there exists $r_0 \in \mathbb{R}^+$ such that $\mathcal{B}_{r_0}(\mathbf{z}_B) \subseteq B_{\mathbf{z}}^*$. Therefore, $B_{\mathbf{z}}^*$ contains an open set and has non-zero probability. Second, it is by definition that $B_{\mathbf{z}}^*$ cannot be expressed as $A_{\mathbf{z}_x} \times \mathcal{Z}_{tx} \times \mathcal{Z}_t$ for any $A_{\mathbf{z}_x} \subset \mathcal{Z}_x$.

Therefore, condition (ii) from the theorem indicates that there exists $\mathbf{t}_1^*, \mathbf{t}_2^*, \mathbf{x}^*$, such that

$$T_2 = \int_{[\mathbf{z}_x \mathbf{z}_t^+] \in B_{\mathbf{z}}^*} p_{\mathbf{z}_x|\mathbf{x}}\left(\mathbf{z}_{\mathbf{x}} \mid \mathbf{x}^*\right) \left(p_{\mathbf{z}_t^+|\mathbf{t},\mathbf{x}}\left(\mathbf{z}_t^+ \mid \mathbf{x}^*, \mathbf{t}_1^*\right) - p_{\mathbf{z}_t^+|\mathbf{t},\mathbf{x}}\left(\mathbf{z}_t^+ \mid \mathbf{x}^*, \mathbf{t}_2^*\right)\right) d\mathbf{z}_x d\mathbf{z}_t^+ \neq 0. \tag{62}$$

Therefore, for such $S^*$, we would have $T_1 + T_2 \neq 0$ which leads to contradiction with Equation 53. We have proved by contradiction that Statement 1 from Step 2 holds, that is, $\bar{h}_x$ does not depend on the treatment variable and interaction variable $\mathbf{z}_t$, $\mathbf{z}_{tx}$.

**Step 4 (Block-wise identifiability of $\mathbf{z}_t$ and $\mathbf{z}_x$).** With the knowledge that $h_x$ does not depend on $\mathbf{z}_t^+$, we now show that there exists an invertible mapping between the true content variable $\mathbf{z}_x$ and the estimated version $\hat{\mathbf{z}}_x$.

As $\bar{h}$ is smooth over $\mathcal{Z}$, its Jacobian can written as:

$$\mathbf{J}_h = \left[\begin{array}{c|c|c} \mathbf{A} := \frac{\partial \mathbf{z}_x}{\partial \hat{\mathbf{z}}_x} & \mathbf{B} := \frac{\partial \mathbf{z}_x}{\partial \hat{\mathbf{z}}_{tx}} & \mathbf{C} := \frac{\partial \mathbf{z}_x}{\partial \hat{\mathbf{z}}_t} \\ \hline \mathbf{D} := \frac{\partial \mathbf{z}_{tx}}{\partial \hat{\mathbf{z}}_x} & \mathbf{E} := \frac{\partial \mathbf{z}_{tx}}{\partial \hat{\mathbf{z}}_{tx}} & \mathbf{F} := \frac{\partial \mathbf{z}_{tx}}{\partial \hat{\mathbf{z}}_t} \\ \hline \mathbf{G} := \frac{\partial \mathbf{z}_t}{\partial \hat{\mathbf{z}}_x} & \mathbf{H} := \frac{\partial \mathbf{z}_t}{\partial \hat{\mathbf{z}}_{tx}} & \mathbf{I} := \frac{\partial \mathbf{z}_t}{\partial \hat{\mathbf{z}}_t} \end{array}\right] \tag{63}$$

where we use notation $\hat{\mathbf{z}}_x = \bar{h}(\mathbf{z})_{1:n_x}$ and $\hat{\mathbf{z}}_{tx} = \bar{h}(\mathbf{z})_{n_x+1:n_x+n_{tx}}, \hat{\mathbf{z}}_t = \bar{h}(\mathbf{z})_{n_x+n_{tx}+1:n}$.

First, we notice that under the condition of Theorem 4.4, there is an invertible mapping between $\mathbf{z}_{tx}$ and $\hat{\mathbf{z}}_{tx}$. Therefore, it must be that $\mathbf{D} = \mathbf{F} = \mathbf{0}$, and that $\mathbf{E}$ is non-singular. Additionally, we have just shown that $\hat{\mathbf{z}}_x$ does not depend on the treatment-related variables $\mathbf{z}_t^+$. Therefore, it follows $\mathbf{B} = \mathbf{C} = \mathbf{0}$. On the other hand, as $h$ is invertible over $\mathcal{Z}$, $\mathbf{J}_{\bar{h}}$ is non-singular. Therefore, $\mathbf{A}$ must be non-singular due to $\mathbf{B} = \mathbf{C} = \mathbf{0}$. Relying on analogous assumptions to prove the invariance of $\hat{\mathbf{z}}_t$ with respect to $\mathbf{z}_x^+$, it follows that $\mathbf{G} = \mathbf{H} = \mathbf{0}$, and that $\mathbf{I}$ must be non-singular.

We note that $\mathbf{A}$ is the Jacobian of the function $\bar{h}_x'(\mathbf{z}_x) := \bar{h}_x(\mathbf{z}) : \mathcal{Z}_x \to \mathcal{Z}_x$, which takes only the covariates part $\mathbf{z}_x$ of the input $\mathbf{z}$ into $\bar{h}_x$. Together with the invertibility of $\bar{h}$, we can conclude that $\bar{h}_x'$ is invertible. Therefore, there exists an invertible function $\bar{h}_x'$ between the estimated and the true $\hat{\mathbf{z}}_x = \bar{h}_x'(\mathbf{z}_x)$, which concludes the proof that $\mathbf{z}_x$ is block-identifiable. Similarly, we are able to conclude $\mathbf{z}_t$ is block-identifiable.

$\square$

# B Derivation of the evidence lower bound

We now introduce the classical derivations of the celebrated evidence lower bound for our generative model. The evidence is the logarithm of the marginal data probability, and we calculate it by weighting it against the variational distribution:

$$\log p(\mathbf{y} \mid \mathbf{x}, \mathbf{t}) = \log \mathbb{E}_{q_\phi(\mathbf{z}_t, \mathbf{z}_{tx}, \mathbf{z}_x \mid \mathbf{y}, \mathbf{t}, \mathbf{x})} \left(\frac{p_\theta(\mathbf{y}, \mathbf{z}_t, \mathbf{z}_{tx}, \mathbf{z}_x \mid \mathbf{t}, \mathbf{x})}{q_\phi(\mathbf{z}_t, \mathbf{z}_{tx}, \mathbf{z}_x \mid \mathbf{y}, \mathbf{t}, \mathbf{x})}\right). \tag{64}$$

We apply Jensen's inequality using the fact that the logarithm is a concave function:

$$\log p(\mathbf{y} \mid \mathbf{x}, \mathbf{t}) \geq \mathbb{E}_{q_\phi(\mathbf{z}_t, \mathbf{z}_{tx}, \mathbf{z}_x \mid \mathbf{y}, \mathbf{t}, \mathbf{x})} \log \left(\frac{p_\theta(\mathbf{y}, \mathbf{z}_t, \mathbf{z}_{tx}, \mathbf{z}_x \mid \mathbf{t}, \mathbf{x})}{q_\phi(\mathbf{z}_t, \mathbf{z}_{tx}, \mathbf{z}_x \mid \mathbf{y}, \mathbf{t}, \mathbf{x})}\right). \tag{65}$$

**Algorithm 1** Training of FCR

**Input:** $\mathbf{y}, \mathbf{y}^0$, shared $\mathbf{x}$, and $\mathbf{t}, \mathbf{t}^0 = 0$
**Output:** $p, q, \mathbf{z}_x, \mathbf{z}_{tx}, \mathbf{z}_t$
**while** Not converged **do**
  *Minimization Stage*
  1. Estimate $p(\mathbf{z}_t \mid \mathbf{t})$, $p(\mathbf{z}_x \mid \mathbf{x})$, $p(\mathbf{z}_{tx} \mid \mathbf{t}, \mathbf{x})$,
      and $p(\mathbf{z}_t^0 \mid \mathbf{t}^0)$, $p(\mathbf{z}_x^0 \mid \mathbf{x})$, $p(\mathbf{z}_{tx}^0 \mid \mathbf{t}^0, \mathbf{x})$
  2. Estimate $q(\mathbf{z}_t, \mathbf{z}_x, \mathbf{z}_{tx} \mid \mathbf{t}, \mathbf{x}, \mathbf{y})$,
  $q(\mathbf{z}_t^0, \mathbf{z}_x^0, \mathbf{z}_{tx}^0 \mid \mathbf{t}^0, \mathbf{x}, \mathbf{y})$
  3. Calculate Kullback-Leibler divergence terms and predict $\hat{\mathbf{t}}$
  4. Calculate similarities $\{\mathbf{z}_x^0, \mathbf{z}_x\}$, $\{\mathbf{z}_t^0, \mathbf{z}_t\}$
  5. Permute $\mathbf{z}_{tx}$ to get $\hat{\mathbf{z}}_{tx}$ and predict permutation labels $\hat{l}$
  6. Minimize $\mathcal{L}_{\text{ELBO}}, \mathcal{L}_{\text{sim}}, \mathcal{L}_{\text{ct}}, \mathcal{L}_{\text{dist}_\mathbf{x}}, \mathcal{L}_{\text{dist}_\mathbf{t}}$
  *Maximization Stage*
  1. Estimate $q(\mathbf{z}_t, \mathbf{z}_x, \mathbf{z}_{tx} \mid \mathbf{t}, \mathbf{x}, \mathbf{y})$,
  2. Permute $\mathbf{z}_{tx}$ to get $\hat{\mathbf{z}}_{tx}$ and predict permutation labels $\hat{l}$
  3. Maximize $\mathcal{L}_{\text{dist}_\mathbf{x}}, \mathcal{L}_{\text{dist}_\mathbf{t}}$
**end while**

Then, we use the factorization of our generative model:

$$\log p(\mathbf{y} \mid \mathbf{x}, \mathbf{t}) \geq \mathcal{L}_{\text{ELBO}} := \mathbb{E}_{q_\phi(\mathbf{z}_t, \mathbf{z}_{tx}, \mathbf{z}_x \mid \mathbf{y}, \mathbf{t}, \mathbf{x})} \log \frac{p_\theta(\mathbf{y} \mid \mathbf{z}_t, \mathbf{z}_{tx}, \mathbf{z}_x) p_\theta(\mathbf{z}_t, \mathbf{z}_{tx}, \mathbf{z}_x \mid \mathbf{t}, \mathbf{x})}{q_\phi(\mathbf{z}_t, \mathbf{z}_{tx}, \mathbf{z}_x \mid \mathbf{y}, \mathbf{t}, \mathbf{x})}. \quad (66)$$

Naming the right hand size of Equation 66 as $\mathcal{L}_{\text{ELBO}}$, we notice that $\mathcal{L}_{\text{ELBO}}$ can be written as the following difference:

$$\mathcal{L}_{\text{ELBO}} = \mathbb{E}_{\bar{q}_\phi} \log p_\theta(\mathbf{y} \mid \mathbf{z}_t, \mathbf{z}_{tx}, \mathbf{z}_x) - \mathbb{E}_{\bar{q}_\phi} \log \frac{q_\phi(\mathbf{z}_t, \mathbf{z}_{tx}, \mathbf{z}_x \mid \mathbf{y}, \mathbf{t}, \mathbf{x})}{p_\theta(\mathbf{z}_t, \mathbf{z}_{tx}, \mathbf{z}_x \mid \mathbf{t}, \mathbf{x})}, \quad (67)$$

where the first term is the reconstruction loss, and the second term is the Kullback-Leibler divergence between the approximate posterior $\bar{q}_\phi = q_\phi(\mathbf{z}_t, \mathbf{z}_{tx}, \mathbf{z}_x \mid \mathbf{y}, \mathbf{t}, \mathbf{x})$ and the prior $p_\theta(\mathbf{z}_t, \mathbf{z}_{tx}, \mathbf{z}_x \mid \mathbf{t}, \mathbf{x})$. Further decomposing this term using the factorization of the generative model and the inference model, we obtain:

$$\begin{aligned}
\mathbb{E}_{\bar{q}_\phi} \log \frac{q_\phi(\mathbf{z}_t, \mathbf{z}_{tx}, \mathbf{z}_x \mid \mathbf{y}, \mathbf{t}, \mathbf{x})}{p_\theta(\mathbf{z}_t, \mathbf{z}_{tx}, \mathbf{z}_x \mid \mathbf{t}, \mathbf{x})} = {} & \mathbb{E}_{q_\phi(\mathbf{z}_t \mid \mathbf{y}, \mathbf{t})} \log \frac{q_\phi(\mathbf{z}_t \mid \mathbf{y}, \mathbf{t})}{p_\theta(\mathbf{z}_t \mid \mathbf{t})} \\
& + \mathbb{E}_{q_\phi(\mathbf{z}_{tx} \mid \mathbf{y}, \mathbf{t}, \mathbf{x})} \log \frac{q_\phi(\mathbf{z}_{tx} \mid \mathbf{y}, \mathbf{t})}{p_\theta(\mathbf{z}_{tx} \mid \mathbf{t}, \mathbf{x})} \\
& + \mathbb{E}_{q_\phi(\mathbf{z}_x \mid \mathbf{y}, \mathbf{x})} \log \left( \frac{q_\phi(\mathbf{z}_x \mid \mathbf{y}, \mathbf{x})}{p_\theta(\mathbf{z}_x \mid \mathbf{x})} \right)
\end{aligned} \quad (68)$$

Finally, recognizing three Kullback-Leibler divergence terms in the right hand side of Equation 68, and injecting this expression into the evidence lower bound expression of Equation 66, we obtain the desired expression:

$$\begin{aligned}
\mathcal{L}_{\text{ELBO}} = {} & \mathbb{E}_{q_\phi(\mathbf{z}_x, \mathbf{z}_t, \mathbf{z}_{tx} \mid \mathbf{x}, \mathbf{t}, \mathbf{y})} \log p_\theta(\mathbf{y} \mid \mathbf{z}_x, \mathbf{z}_t, \mathbf{z}_{tx}) \\
& - D_{KL}(q_\phi(\mathbf{z}_x \mid \mathbf{x}, \mathbf{y}) \| p_\theta(\mathbf{z}_x \mid \mathbf{x})) \\
& - D_{KL}(q_\phi(\mathbf{z}_t \mid \mathbf{t}, \mathbf{y}) \| p_\theta(\mathbf{z}_t \mid \mathbf{t})) \\
& - D_{KL}(q_\phi(\mathbf{z}_{tx} \mid \mathbf{t}, \mathbf{x}, \mathbf{y}) \| p_\theta(\mathbf{z}_{tx} \mid \mathbf{t}, \mathbf{x})),
\end{aligned} \quad (69)$$

# C Training Details

We provide additional training information in this section, including a detailed algorithm for the training process of FCR (Algorithm 1).

# D  Hyperparameter selection

We split the data into four datasets: train/validation/test/prediction, following the setup from previous works (Lotfollahi et al., 2023; Wu et al., 2023). First we hold out the 20% of the control cells for the final cellular prediction tasks (prediction). Second, we hold 20% of the rest of data for the task of clustering and statistical test (test). Third, the data excluding the prediction and clustering/test sets are split into training and validation sets with a four-to-one ratio.

For the hyperparameter tuning procedure, conduct the exhaustive hyperparameter grid search with n_epoch=100 on the loss assessed on the validation data. The hyperparameter search space is shown in Table 2.

| Parameter | Values |
|---|---|
| $\omega_1$ | {0.5, 1.0, 2.0, 3.0, 4.0, 5.0, 6.0, 7.0, 8.0, 9.0, 10.0} |
| $\omega_2$ | {0.5, 1.0, 2.0, 3.0, 4.0, 5.0, 6.0, 7.0, 8.0, 9.0, 10.0} |
| $\omega_3$ | {0.1, 0.3, 0.5, 0.7, 0.9, 1.0, 3.0, 5.0, 7.0, 10.0} |

Table 2: Hyperparameter space

# E  Datasets and Preprocessing

In this section, we provide detailed descriptions of the four datasets and their corresponding pre-processing procedures. These datasets contain cells from many cell lines that are described in Table 3.

| No. | Cell Line | Origin |
|---|---|---|
| 1 | IALM | Lung |
| 2 | SKMEL2 | Skin |
| 3 | SH10TC | Stomach |
| 4 | SQ1 | Lung |
| 5 | BICR31 | Upper Aerodigestive Tract |
| 6 | DKMG | Central Nervous System |
| 7 | BT474 | Breast |
| 8 | TEN | Endometrium |
| 9 | COLO680N | Oesophagus |
| 10 | CAOV3 | Ovary |
| 11 | SKMEL3 | Skin |
| 12 | NCIH226 | Lung |
| 13 | LNCAPCLONEFGC | Prostate |
| 14 | RCC10RGB | Kidney |
| 15 | BICR6 | Upper Aerodigestive Tract |
| 16 | BT549 | Breast |
| 17 | CCFSTTG1 | Central Nervous System |
| 18 | RERFLCAD1 | Lung |
| 19 | UMUC1 | Urinary Tract |
| 20 | RCM1 | Large Intestine |
| 21 | LS1034 | Large Intestine |
| 22 | SNU1079 | Biliary Tract |
| 23 | NCIH2347 | Lung |
| 24 | COV434 | Ovary |

Table 3: Cell line information for multiplex experiments

## E.1  SciPlex dataset

This dataset includes three cancer cell lines exposed to 188 different compounds. In total, this experiment profiled approximately 650,000 single-cell transcriptomes across roughly 5,000 independent samples (Srivatsan et al., 2020). For our experiments, we selected only the HDAC inhibitors that

were shown to be effective in these three cell lines (Srivatsan et al., 2020). The following is the list of HDAC inhibitors used in Table 4.

| HDAC Inhibitor Drugs |
| --- |
| Belinostat |
| Mocetinostat |
| Panobinostat |
| Pracinostat |
| Dacinostat |
| Quisinostat |
| Tucidinostat |
| Givinostat |
| AR-42 |
| Tacedinaline |
| CUDC-907 |
| M344 |
| Resminostat |
| Entinostat |
| TSA |
| CUDC-101 |

Table 4: The list of HDAC inhibitors

We extracted the cells treated with HDAC inhibitors along with their corresponding control groups. After filtering out low-quality cells, we normalized the raw counts. From these, we retained the top 5,000 highly expressed genes. Ultimately, we analyzed 90,462 cells, including both treated and control groups. Cell lines and repetition numbers were used as covariates, with treatment dosage designated as the treatment variable.

### E.2 MultiPlex-Tram dataset

This dataset is referred to as experiment-5 in the paper McFarland et al. (2020). It contains in total 20,028 Trametinib treated cells, DSMO control cells, and 24 cell lines (McFarland et al., 2020). The cells are treated with 100nM Trametinib for 3, 6, 12, 24, 48 hours respectively. We removed the low quality cells and normalize the raw counts. Next, we kept first 5,000 differentially expressed genes. Finally, we have 13,713 cells in total for the experiments and down streaming evaluation.

### E.3 Multiplex-7 dataset

This dataset is labeled as experiment-3 in McFarland et al. (2020), includes 72,326 cells treated with seven different compounds across 24 cell lines. Details of the seven treatments are provided in Table 5. Following the removal of low-quality cells and normalization of raw counts, we retained the top 5,000 differentially expressed genes, yielding 61,552 cells for the experiments and downstream analysis.

| Drug | Hours |
| --- | --- |
| DMSO | 6 hours |
| DMSO | 24 hours |
| BRD3379 | 6 hours |
| BRD3379 | 24 hours |
| Dabrafenib | 24 hours |
| Navitoclax | 24 hours |
| Trametinib | 24 hours |

Table 5: The multiPlex-7 dataset's treatments

### E.4 Multiplex-9 dataset

referred to as experiment-10 in McFarland et al. (2020), consists of 37,856 cells across 24 cell lines, treated with 9 drugs (including a control). The list of drugs can be found in Table 6. After filtering out low-quality cells and normalizing the raw counts, we retained the top 5,000 differentially expressed genes. This resulted in a total of 19,524 cells for the experiments and downstream evaluation.

| Drug | Hours |
|------|-------|
| DMSO | 24 hours |
| Everolimus | 24 hours |
| Afatinib | 24 hours |
| Taselisib | 24 hours |
| AZD5591 | 24 hours |
| JQ1 | 24 hours |
| Gemcitabine | 24 hours |
| Trametinib | 24 hours |
| Prexasertib | 24 hours |

Table 6: Drugs list of the multiPlex-9 dataset

## F  Experimental Setups and Additional Results

### F.1  Training Details

In this subsection, we layout the training parameters for each datasets as follows.

**sciPlex**   For the sciPlex datasets, the dimensions are as follows: $\mathbf{z}_x$ is 32, $\mathbf{z}_{tx}$ is 64, and $\mathbf{z}_t$ is 32. Additionally, we set the hyperparameters to $\omega_1 = 3.0$, $\omega_2 = 3.0$, and $\omega_3 = 5.0$, with a batch size of 2046. Note that in sciPlex, we treat the dosages of the HDAC inhibitors as the treatment variable.

**multiPlex-Tram**   or the multiPlex-Tram dataset, the dimensions are set as follows: $\mathbf{z}_x$ is 32, $\mathbf{z}_{tx}$ is 32, and $\mathbf{z}_t$ is 32. The hyperparameters are $\omega_1 = 5.0$, $\omega_2 = 5.0$, $\omega_3 = 1.0$, with a batch size of 2046. In this dataset, Trametinib treatment time is considered as the treatment variable.

**multiPlex-7**   For the multiPlex-7 dataset, the dimensions are $\mathbf{z}_x = 32$, $\mathbf{z}_{tx} = 64$, and $\mathbf{z}_t = 32$. The hyperparameters are $\omega_1 = 1.0$, $\omega_2 = 0.5$, and $\omega_3 = 0.1$, with a batch size of 2046.

**multiPlex-9**   For the multiPlex-9 dataset, the dimensions are $\mathbf{z}_x = 32$, $\mathbf{z}_{tx} = 64$, and $\mathbf{z}_t = 32$. The hyperparameters are $\omega_1 = 1.0$, $\omega_2 = 0.5$, and $\omega_3 = 0.1$, with a batch size of 2046.

Additionally, the autoencoder learning rate is set to $3 \times 10^{-4}$, the discriminator learning rates are also $3 \times 10^{-4}$, and the number of discriminator training steps is 10.

### F.2  Simulation Study

We provide some empirical assessment of our identifiability theory using simulations.

**Data Generation**   Following the simulation protocol outlined in Kong et al. (2022), Khemakhem et al. (2020) and Lopez et al. (2023), we simplify the simulation setup by setting the dimensions of $\mathbf{z}_x$ and $\mathbf{z}_t$ to 1, while $\mathbf{z}_{xt}$ has a dimension of 4. Specifically, we use a sample size of 5,000 and define our variables as follows:

$$\mathbf{t} \sim \text{Unif}(\{1, 2, 3\}) \tag{70}$$

Here, $\mathbf{t}$ represents the treatment variable, uniformly distributed over three discrete values.

$$\mathbf{x} \sim \text{Unif}(\{100, 1000, 5000\}) \tag{71}$$

$\mathbf{x}$ denotes the covariate, also uniformly distributed but over a wider range of values.

$$\mathbf{z}_x \sim \text{Normal}(\mathbf{x}/2, 1) \tag{72}$$

$\mathbf{z}_x$ is the latent variable associated with $\mathbf{x}$, following a normal distribution with mean $\mathbf{x}/2$ and unit variance.

$$\mathbf{z}_t \sim \text{Normal}(\mathbf{t}/2, 1) \tag{73}$$

$\mathbf{z}_t$ is the latent variable associated with $\mathbf{t}$, also normally distributed with mean $\mathbf{t}/2$ and unit variance.

$$\mathbf{z}_{tx} \sim \text{Normal}(\mathbf{x} \cdot \mathbf{t}, \mathbf{I}_4) \tag{74}$$

$\mathbf{z}_{tx}$ represents the interaction between $\mathbf{x}$ and $\mathbf{t}$, following a multivariate normal distribution with mean $\mathbf{x} \cdot \mathbf{t}$ and covariance matrix $\mathbf{I}_4$ (the 4-dimensional identity matrix).

Finally, we define our output $\mathbf{y}$ as a function of these latent variables:

$$\mathbf{y} = g(\mathbf{z}_x, \mathbf{z}_{tx}, \mathbf{z}_t) \tag{75}$$

Here, $g$ is implemented as a 2-layer MLP with Leaky-ReLU activation, following the approach of Kong et al. (2022) and Khemakhem et al. (2020). The output $\mathbf{y}$ is a real-valued vector with a dimension of 96.

**Evaluation** To assess the component-wise identifiability of the interaction components, we compute the Mean Correlation Coefficient (MCC) between $\mathbf{z}_{tx}$ and $\hat{\mathbf{z}}_{tx}$. MCC is a standard metric in Independent Component Analysis (ICA) literature, where a higher MCC indicates better identifiability. MCC reaches 1 when latent variables are perfectly identifiable (up to a component-wise transformation). We compute the MCC between the original sources and the corresponding latent variables sampled from the approximate posterior. As for the iVAE evaluation framework, we first calculate the correlation coefficients between all pairs of source and latent components. Then, we solve a linear sum assignment problem to map each latent component to the source component that correlates best with it, effectively reversing any latent space permutations. A high MCC indicates successful identification of the true parameters and recovery of the true sources, up to point-wise transformations. This is a standard performance metric in ICA (Khemakhem et al., 2020).

**Results** Our results suggest that our method, FCR largely outperforms existing variational autoencoder-based approaches in identifying the latent interactive components $\mathbf{z}_{tx}$. As shown in Table 7, FCR achieves an almost perfect Mean Correlation Coefficient (MCC), compared to other methods that have poor performance. Even the iVAE baseline falls short of FCR's capability in recovering the true latent structure. These results indicate that FCR offers a significant advancement in component-wise identifiability for complex, interacting latent variables in causal representation learning.

| Method | MCC |
|---|---|
| FCR | **0.91** $\pm$ 0.03 |
| $\beta$-VAE | 0.38 $\pm$ 0.12 |
| FactorVAE | 0.37 $\pm$ 0.08 |
| iVAE | 0.77 $\pm$ 0.07 |

Table 7: Mean Correlation Coefficient (MCC) of $\mathbf{z}_{tx}$ for different methods

### F.3 Additional Clustering Details and Results

**Clustering Approach** Our clustering analysis utilizes the learned representations $\mathbf{z}_x$, $\mathbf{z}_{tx}$, and $\mathbf{z}_t$ for our method, while all available representations were used for baseline methods. It's important to note that baseline models were trained using their default settings.

We employed the following clustering approach for different scenarios:

1. **Clustering on $\mathbf{x}$:** We applied the Leiden clustering algorithm to the different representations and evaluated the results using the labels of $\mathbf{x}$.

2. **Clustering on** $\mathbf{t}$**:** Similarly, we used the Leiden algorithm on the representations and assessed the outcomes with the labels of $\mathbf{t}$.

3. **Clustering on** $\mathbf{x}$ **combined with** $\mathbf{t}$**:** We ran Leiden clustering on the combined representations and evaluated the results using both the labels of $\mathbf{x}$ and $\mathbf{t}$.

This approach allows us to assess the efficacy of our learned representations in capturing the underlying structure of both individual and combined variables.

**Evaluation Metric**    The evaluation metric, Normalized Mutual Information (NMI), is defined as follows:

$$\text{NMI}(Y, C) = \frac{2 \sum_{k=1}^{K} \sum_{l=1}^{L} p_{kl} \log \left( \frac{p_{kl}}{p_k^Y p_l^C} \right)}{\left( -\sum_{k=1}^{K} p_k^Y \log p_k^Y \right) + \left( -\sum_{l=1}^{L} p_l^C \log p_l^C \right)}, \tag{76}$$

where:

- $Y = \{y_1, \ldots, y_K\}$ denotes the set of class labels,
- $C = \{c_1, \ldots, c_L\}$ denotes the set of cluster labels,
- $p_{kl}$ is the joint probability of a data sample belonging to class $k$ and cluster $l$,
- $p_k^Y$ is the marginal probability of a sample belonging to class $k$,
- $p_l^C$ is the marginal probability of a sample belonging to cluster $l$.

Note that these probabilities are computed from the same dataset, but with respect to the true class labels and the assigned cluster labels, respectively.

**Additional Results**    Additional clustering results for the multiPlex-tram, multiPlex-7, and multiPlex-9 datasets are presented in Figure 5.

### F.4    Statistical Tests and More results

Due to the computational complexity of kernel calculations, we adopted a sampling approach with 2,000 samples, repeating the process 100 times to report the results. For the baseline methods, we sampled their latent spaces to match the dimensions of $\mathbf{z}_x$, $\mathbf{z}_t$, and $\mathbf{z}_{tx}$. This sampling was repeated 20 times, and we reported the best results for comparison with FCR. The test results for the multiPlex-tram, multiPlex-7, and multiPlex-9 datasets are shown in the Figure 6

### F.5    Conditional Cellular Response Prediction

For this task, we use FCR's latent representations to predict gene expression levels and report the corresponding $R^2$ scores. Specifically, our approach enables the prediction of cellular responses at the single-cell level. The focus of this paper is on predicting cellular responses (expression of 2,000 genes) in control cells subjected to drug treatments. Our comparative analysis includes CPA, VCI, sVAE, scGEN, and CINEMA-OT, as these methods are specifically designed for cellular prediction tasks.

We utilize FCR to extract the control's latent representations $[\mathbf{z}_x^0, \mathbf{z}_{tx}^0, \mathbf{z}_t^0]$ and the corresponding experimental representations $[\mathbf{z}_x, \mathbf{z}_{tx}, \mathbf{z}_t]$. The decoder $g$ is then used to predict the gene expression levels as $\hat{\mathbf{y}} = g(\mathbf{z}_x^0, \mathbf{z}_{tx}^0, \mathbf{z}_t^0)$. The $R^2$ score is used to evaluate the predictions, and we sampled 20% of each dataset for testing, repeating this process five times.

For iVAE and VCI, we used treatments, covariates, and gene expression data to learn latent variables and predict gene expression. In contrast, for scVI, $\beta$VAE, and factorVAE, we only used gene expression data to learn latent variables and make predictions.

The $R^2$ score is a key metric for assessing the accuracy of predictive models. It measures the proportion of variance in the dependent variable that is explained by the independent variables. An $R^2$ score of 1 indicates perfect prediction accuracy, meaning all variations in the target variable are

fully explained by the model's inputs, while a score of 0 suggests that the model fails to capture any variance in the target variable.

$$R^2 = 1 - \frac{\sum_{i=1}^{n}(y_i - \hat{y}_i)^2}{\sum_{i=1}^{n}(y_i - \bar{y})^2}, \tag{77}$$

where $y_i$ is the actual values, $\hat{y}_i$ is the predicted values, $\bar{y}$ is the mean of average values, $n$ is the number of observations.

**Additional Metric**   We further utilize the Mean Squared Error (MSE) of the top 20 differentially expressed genes (DEGs) for post-treatment. These 20 genes are selected for showing statistically significant differences in expression levels for each cell line with drug treatments compared to control samples. The same procedures are also carried out in Roohani et al. (2024). Note here, we did not compare with CINEMA-OT and scGEN because they are only for binary treatments.

| Dataset | FCR | VCI | CPA | sVAE |
|---|---|---|---|---|
| sciPlex | **0.12** ± 0.03 | 0.15 ± 0.04 | 0.15 ± 0.04 | 0.16 ± 0.05 |
| multiPlex-tram | **0.10** ± 0.07 | 0.13 ± 0.08 | 0.13 ± 0.08 | 0.14 ± 0.07 |
| multiPlex-7 | **0.18** ± 0.07 | 0.21 ± 0.08 | 0.22 ± 0.08 | 0.23 ± 0.07 |
| multiPlex-9 | 0.24 ± 0.06 | 0.27 ± 0.04 | **0.23** ± 0.06 | 0.26 ± 0.05 |

Table 8: Mean Squared Error (MSE) of top 20 Differentially Expressed Genes (DEGs) for different methods across datasets

### F.6   Ablation Study

We present the ablation study results for the hyperparameters $\omega_1$, $\omega_2$, and $\omega_3$. Initially, we set $\omega_1$, $\omega_2$, and $\omega_3$ to [1,1,1]. Then, we varied each parameter independently to $[1, 3, 5, 10, 20]$, keeping the other parameters fixed at 1. Figure 7 illustrates the NMI scores for clustering on $\mathbf{x}$, $\mathbf{t}$, or both $\mathbf{x}\&\mathbf{t}$. Figure 8 shows the $R^2$ scores for each parameter.

### F.7   Visualization

Visualizing latent representations provides intuitive insights into the distinct characteristics and attributes captured by each representation. Using Uniform Manifold Approximation and Projection (UMAP) (McInnes et al., 2018), we visualized the latent representations $\mathbf{z}_x$, $\mathbf{z}_{tx}$, and $\mathbf{z}_t$ derived from the sciPlex dataset (Figure 9).

From these visualizations, we observe that $\mathbf{z}_x$ effectively captures covariate-specific information, showing clear separation in Figure 9, but it does not reflect treatment information. In contrast, the UMAP visualization of $\mathbf{z}_t$ reveals a clear pattern, where cells treated with higher dosages are positioned at the bottom, and control or lower-dosage treated cells are found at the top. Additionally, various cell lines intermingle within the plot, indicating that $\mathbf{z}_t$ successfully captures drug responses across different cell lines.

Most importantly, Figure 9 demonstrates that $\mathbf{z}_{tx}$ captures cell line-specific treatment responses, representing a balanced integration of both covariate and treatment information. Specifically, in the sciPlex dataset, the MCF7 cell line shows a pronounced cell line-specific response, aligning with findings from biological literature (Srivatsan et al., 2020), while the K562 cell line exhibits a less distinct response. This representation confirms the strong, unique responses in certain cell lines, highlighting the validity and precision of our method in capturing nuanced biological behaviors.

### F.8   Pilot Study On The Unseen Drug Responses

Predicting responses to novel treatments is a pivotal and fast-evolving field in drug discovery. However, the biological literature indicates that cellular responses are highly context-dependent (McFarland et al., 2020). This complexity poses significant challenges for AI-driven drug discovery, which often struggles to achieve success in clinical trials.

Our motivation for developing FCR stems from the need to understand how cellular systems react to treatments and identify conditions that can deepen our understanding of these responses. FCR enables

the analysis of drug interactions with covariates and contextual variables. Additionally, predicting cellular responses to new treatments necessitates prior knowledge, such as chemical structure and molecular function, and comparisons with known treatments. Without this context, predictions can be unreliable.

In this paper, our primary focus is not on predicting responses to unseen treatments. However, given the relevance of this topic, we conducted pilot experiments to showcase the potential future applications of FCR. The multiPlex-Tram and multiPlex-7 datasets share the same cell lines and Trametinib-24 hours treatment, along with other different treatments. By utilizing these two datasets, we established the following experimental settings for unseen prediction scenario:

1. **Drug Hold-out Setup**: We held out two cell lines, ILAM and SKMEL2, from the Multiplex-Tram dataset, which had been treated with Trametinib for 24 hours. We trained a FCR model using the remaining data, and this model is referred to as $M_h$. We denote the dataset (Multiplex-Tram dataset without Trametinib-24h treated ILAM and SKMEL2) as $D^h$

2. **Prior Knowledge Model**: We trained another FCR model, $M_p$, using the Multiplex-7 dataset, which includes the ILAM and SKMEL2 cell lines treated with Trametinib for 24 hours denoted as $D^p$. We treat this model, $M_p$, as a prior knowledge model.

3. **Transfer MLP**: We extract both $\mathbf{z}_{tx}^p$ from $M_p$ and $\mathbf{z}_{tx}^h$ from $M_h$ for $D^p$. Then we trained a 1-layer MLP (the same dimension as $\mathbf{z}_{tx}^p$) to transfer $\mathbf{z}_{tx}^p$ to $\mathbf{z}_{tx}^h$, by minimize the MSE between them.

4. **Contextual Prior Representation**: We extracted the $\mathbf{z}_{tx}^p$ representations from model $M_p$ for ILAM and SKMEL2 in holdout set. Then transfer $\mathbf{z}_{tx}^p$ by the previous MLP to $\hat{\mathbf{z}}_{tx}^h$ as a prior contextual embedding. For the hold-out cell lines ILAM and SKMEL2 in the Multiplex-Tram dataset, we extracted $\mathbf{z}_x^h$ from model $M_h$, and Trametinib-24h $\mathbf{z}_t^h$ from other treated cell lines in $D^h$

5. **Representation Matching**: Then we match the $\hat{\mathbf{z}}_{tx}^h$ by $\mathbf{z}_x^p$ similarity on ILAM and SKMEL2 across Multiplex-tram and Multiplex-7 in the prior knowledge model space.

6. **Prediction**: We predicted the unseen 24 hours Trametinib responses for the ILAM and SKMEL2 cell lines in holdout dataset using the formula: $\hat{\mathbf{y}} = g(\mathbf{z}_x^h, \hat{\mathbf{z}}_{tx}^h, \mathbf{z}_t^h)$, where $\hat{\mathbf{z}}_{tx}^h$ is the corresponding matched prior contextual representation, $\mathbf{z}_t^h$ is the tramnib-24 average value in other cell lines.

7. **Evaluation**: We computed the $R^2$ and MSE for the top 20 differentially expressed genes (DEGs) based on the predicted values and compared these results with those from the VCI model. The paper's OOD prediction setups.

| Method | $R^2$ | MSE |
|--------|-------|-----|
| **FCR** | $\mathbf{0.74} \pm 0.03$ | $\mathbf{0.52} \pm 0.11$ |
| **VCI** | $0.71 \pm 0.05$ | $0.55 \pm 0.08$ |

Table 9: Out-of-Distribution (OOD) Performance Results

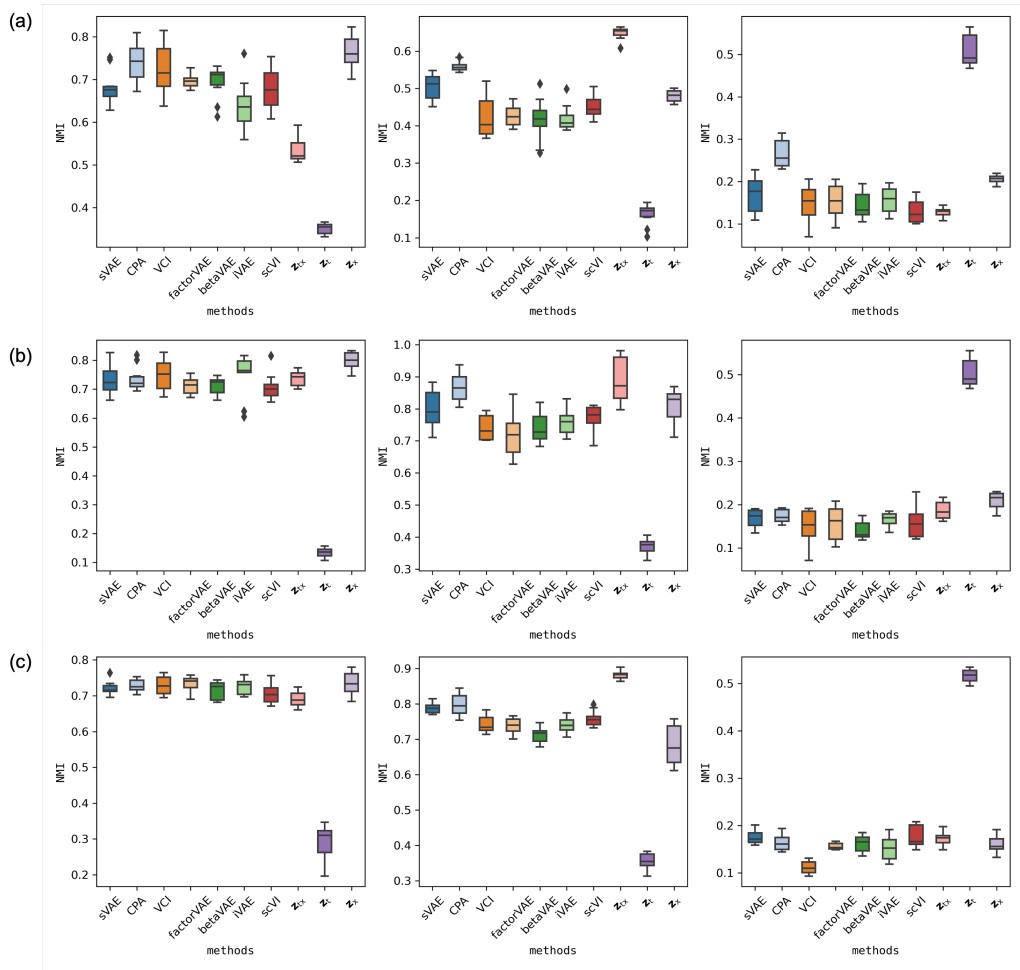

Figure 5: The clustering results. (a) For the multiPlex-Tram dataset, the first column shows the NMI value for clustering on $\mathbf{x}$, the second column shows the NMI value for clustering on $\mathbf{xt}$, and the third column shows the NMI value for clustering on $\mathbf{t}$.(b) For the multiPlex-5 dataset, the first column presents the NMI value for clustering on $\mathbf{x}$, the second column shows the NMI value for clustering on $\mathbf{xt}$, and the third column shows the NMI value for clustering on $\mathbf{t}$. (c) For the multiPlex-9 dataset, the first column displays the NMI value for clustering on $\mathbf{x}$, the second column shows the NMI value for clustering on $\mathbf{xt}$, and the third column shows the NMI value for clustering on $\mathbf{t}$.

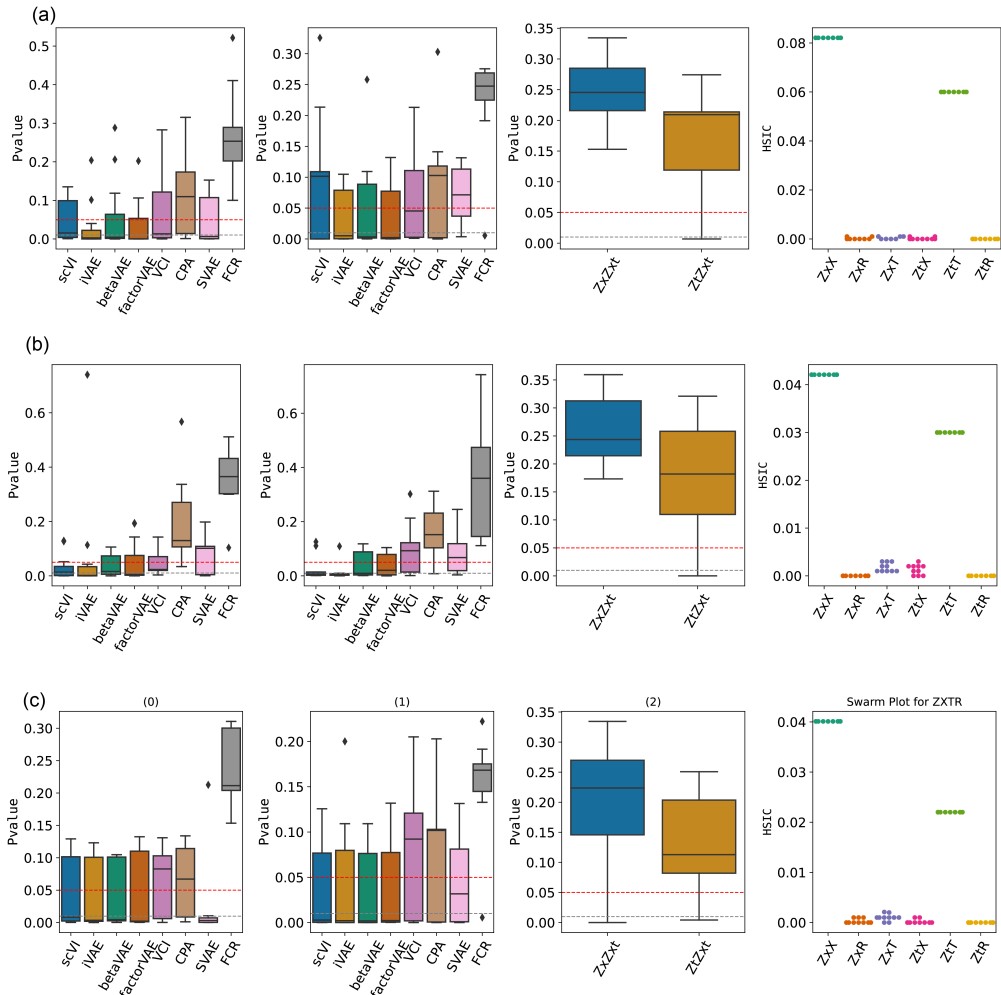

Figure 6: Conditional independence test results for the multiPlex-7 (a), multiPlex-tram (b), and multiPlex-9 (c) datasets. The first column presents the p-values for the conditional independence test of $\mathbf{z}_x$ and $\mathbf{t}$ conditioned on $\mathbf{x}$, with the red dashed line indicating a significance threshold of 0.05. The second column shows the p-values for the conditional independence test of $\mathbf{z}_t$ and $\mathbf{x}$ conditioned on $\mathbf{t}$. The third column presents the p-values for the conditional independence tests of $\mathbf{z}_x$ and $\mathbf{z}_{tx}$ conditioned on $\mathbf{x}$, and of $\mathbf{z}_t$ and $\mathbf{z}_{tx}$ conditioned on $\mathbf{t}$. The fourth column shows the HSIC values of $\mathbf{z}_x$ with $\mathbf{x}$, $\mathbf{t}$, and random variables (R), as well as the HSIC values of $\mathbf{z}_t$ with $\mathbf{x}$, $\mathbf{t}$, and random variables (R).

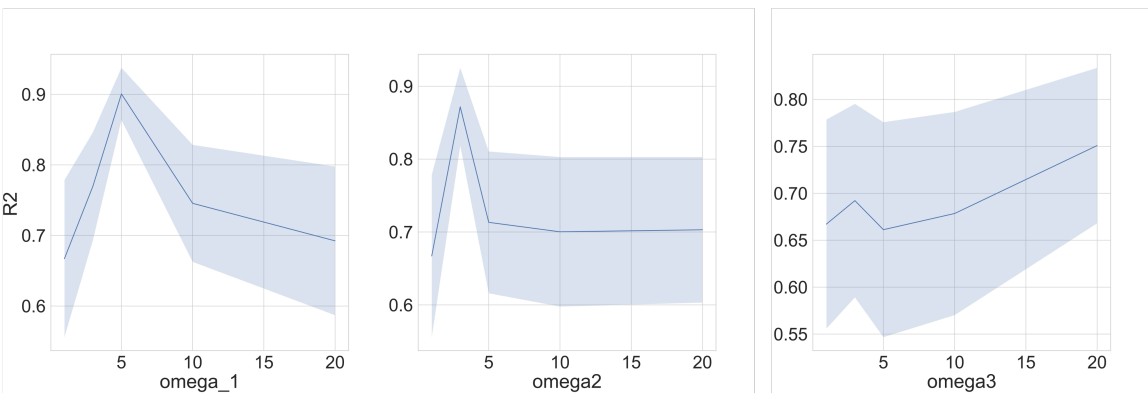

Figure 7: Clustering Ablation Results

Figure 8: $R^2$ score with different $\omega_1$, $\omega_2$ and $\omega_3$ values

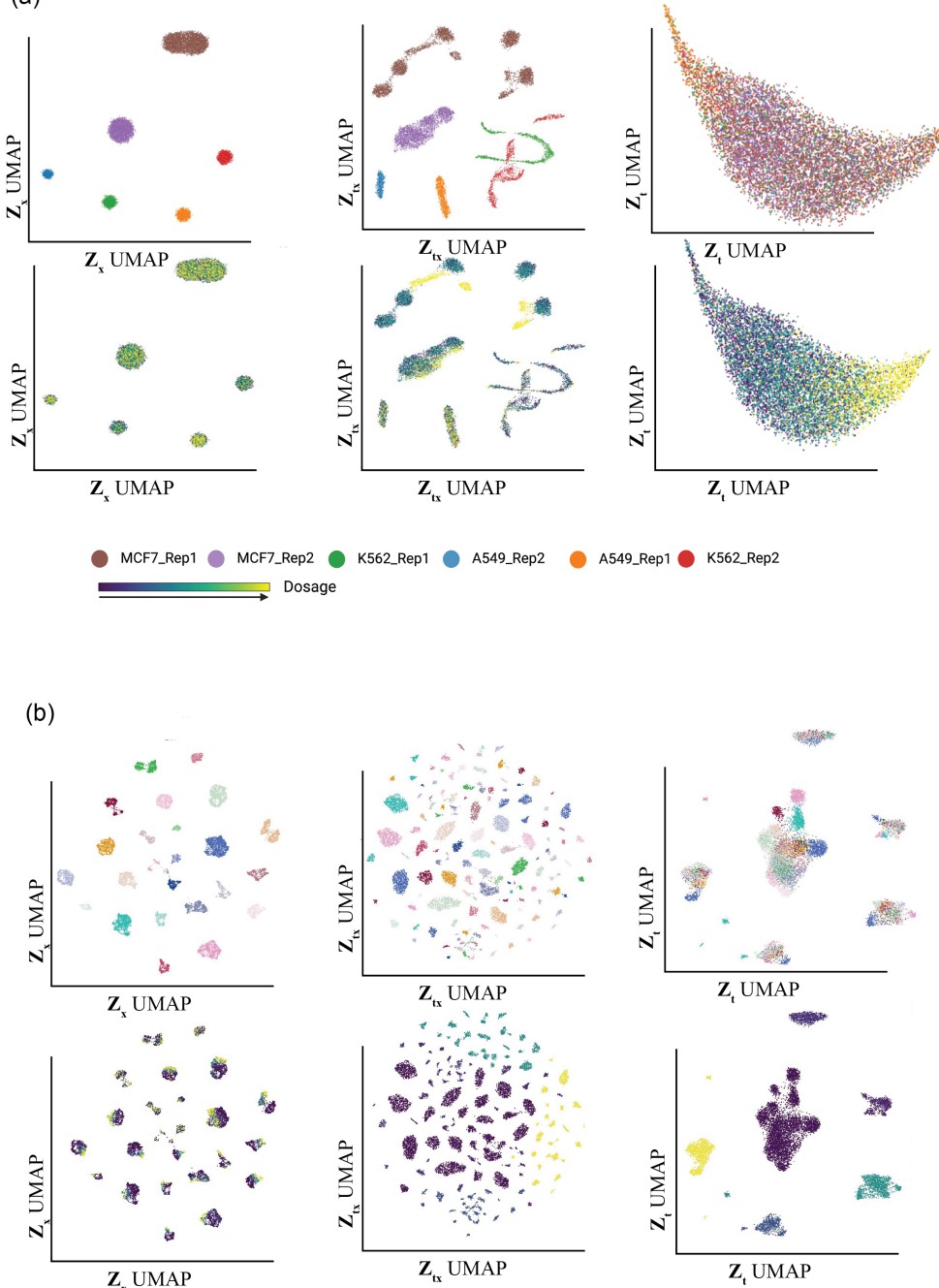

Figure 9: (a) UMAP visualization of the sciPlex dataset. The first row distinguishes the data by cell type, while the second row distinguishes by treatment time, with brighter colors indicating longer treatment durations. (b) UMAP visualization of the multiPlex-tram dataset. The first row distinguishes by cell type, and the second row distinguishes by treatment time, with brighter colors representing longer treatment durations.

