# OpenReview forum: "Learning Identifiable Factorized Causal Representations of Cellular Responses"
_NeurIPS.cc/2024/Conference — NeurIPS 2024 poster_

### Official Review · Reviewer_WECd · 2024-07-11

**Soundness:** 3
**Presentation:** 2
**Contribution:** 3
**Rating:** 5
**Confidence:** 2

**Summary:**

This paper proposes FCR, a causal VAE-based model that aims to decompose and cluster covariates, treatments, and their interactions in the latent space.

**Strengths:**

This paper has a solid and rigorous mathematical foundation, and the assumptions are validated after the model is trained.

**Weaknesses:**

This paper is limited in readability, mainly because of word redundancy, making it hard for the reader to follow the authors’ ideas. Besides, Figure 2 outlines the FCR model, but the encoders and decoders are disconnected, so it is not intuitive to understand how we train components such as $p(z_t|t)$ and $p(z_x|t)$.

**Questions:**

Is it possible to enhance other model’s interpretability using the factorized $z_x,z_t,z_{tx}$?

**Limitations:**

I cannot find a section discussing the limitations of this work, but I believe the noise and hidden variables in the single-cell data will cap the model performance.

---

> ### Author Rebuttal · Authors · 2024-08-06
>
> $\textbf{Readability}$:
>
> Thank you for your feedback regarding the readability of our manuscript. We appreciate your honest assessment and understand the importance of clear and concise writing. We will carefully review the manuscript to eliminate any word redundancy and improve the overall clarity, ensuring that our ideas are communicated more effectively.
>
> $\textbf{$p(\mathbf{z}_t|\mathbf{t})$ and $p(\mathbf{z}_x|\mathbf{x})$ learning}$:
>
> Thank you for this question. I’d like to clarify that both $x$ and $t$ are inputs to the neural network, which then encodes them to learn the prior mean and standard deviation for $z_t$ and $z_x$. We use the reparameterization trick to sample $p(z_t|t)$ and $p(z_x|x)$. We will add this explanation under the model figure in the manuscript to provide clearer context.
>
> $\textbf{Interpretability}$:
>
>
> There are generally two approaches to interpreting latent representations, which we also plan to explore in future work:
> 1. **Latent Traversal**: Since $z_{tx}$ is element-wise identifiable, we can use latent traversal across all dimensions of $z_{tx}$. This involves fixing all latent variables $z_{tx}^i$ to their inferred values for each cell's gene expression and then varying the value of one latent at a time to observe the reconstructed gene expression. This method allows us to see how each dimension of $z_{tx}$ influences gene expression levels.
> 2. **Interpretable Decoder System**: We can also employ an interpretable decoder, such as LDVAE [1], which uses a linear decoder. By analyzing the weights of the latent representations assigned to each gene, we can interpret how different representations influence each gene.
> 3. **On-going Work**: In light of the neural additive model [2], we are developing a non-linear interpretable encoder as an extension of LDVAE. Specifically, for each latent dimension, there is a sub-neural network to capture information specific to each latent dimension. A logistic regression layer is added to the end of the neural network. So, by analyzing the logistic regression weights (loading matrix), we can conclude how much one latent variable contributes to the reconstruction of a specific gene. **An illustration of this model can be found in the rebuttal PDF file Figure D**.
>
>
>
> $\textbf{Limitation discussions}$:
>
> Thank you for raising this important question. Indeed, this paper has some limitations that we plan to address in future work.
>
>
> Firstly, the generating function 𝑔 used in our model is either deterministic or incorporates Gaussian noise. Future research will investigate more general scenarios with stochastic generating functions, which are more representative of real-world applications. Secondly, while this paper does not include interpretability mechanisms for FCR, we have proposed potential approaches to enhance interpretability, as discussed in the previous section. Lastly, due to the limitations of the available public datasets, we recognize that testing FCR on datasets with a broader range of drugs and more complex covariate scenarios will be an intriguing direction for future work.
>
>
> [1] Svensson, V., et al. (2020). Interpretable factor models of single-cell RNA-seq via variational autoencoders. Bioinformatics (Oxford, England), 36(11), 3418–3421.
> [2] Agarwal, R., et al. (2021). Neural additive models: Interpretable machine learning with neural nets. Advances in neural information processing systems, 34, 4699-4711.

---

> > ### Comment · Reviewer_WECd · 2024-08-12
> >
> > Thank you for the authors' reply addressing much of my concern. I will keep my score unchanged.

---

> ### Author Response · Authors · 2024-08-13
>
> Thank you for reading our response. Please feel free to let us know if there is anything needs to be clarified after the rebuttal. We value your feedback and want to address your remaining concerns.

---

### Official Review · Reviewer_q35E · 2024-07-12

**Soundness:** 3
**Presentation:** 2
**Contribution:** 3
**Rating:** 6
**Confidence:** 3

**Summary:**

The authors propose a method for disentangling into factorized representations dependent on only covariates, only treatment and the interaction between treatment and covariates. Introducing a set of assumptions, the authors prove the identifiability of these disentangled variables based on previous proofs on non-linear ICA. The authors implement the model using variational inference and a set of regularizers and discriminators to ensure the needed independence and disentangling. The authors validate the clustering performance, conditional independencies and conditional cellular response prediction against commonly used methods for perturb-seq prediction and show generally very good performance.

**Strengths:**

- Non-linear interactions between covariates and treatment are very likely and have to be modeled
- Given a set of additional assumptions, the method has some guarantees of disentangling, which is also shown empirically
- Understanding (e.g. through clustering) interactions between covariates and treatment is important for precision medicine applications
- Good performance as shown by authors

**Weaknesses:**

- Complicated implementation with many loss functions and minmax training, a discussion on how easy to train the method is would be good (dependence of performance on hyperparameters and hyperparameters on data set).
- The ablation study should include a comparison without the different loss terms to be able to see the importance of the different parts of the model
- Assumption 4.3 could be quite strong in many settings, if for every covariate every drug response has to be available. A comment on sufficient experiments would be good and how common this is.
- For some of the assumptions (e.g. substantive changes of the distribution) empirically arguing that this is the case in common data sets would make the real world application of the proof (instead of the purely academic pursuit) more convincing.

**Questions:**

- How were the hyperparameters chosen for the different data sets?
- Maybe I am misunderstanding "control", but in the conditional cellular response prediction, why is it necessary to use $z_x^0$ instead of $z_x$ in the function $g$ if $z_x$ is independent of $t$?
- For prediction validation, what is train/val/test protocol? Is the correlation of y in the train set?

**Limitations:**

The authors could discuss the limitations (based on assumptions) more directly.

---

> ### Author Rebuttal · Authors · 2024-08-06
>
> $\textbf{Hyperparameter selection}$:
> Thank you for raising this important question, we appreciate that it is a critical point for evaluating how well the method works. Please check **General Response to All Reviewer b**.
>
> $\textbf{Ablation studies}$:
>
> In addition, as the reviewer suggested,  we did the components ablation study (without the different loss terms) on the clustering task (sciPlex dataset) with the NMI score reported.
> $\textbf{Table 1: Benchmark on $\mathbf{z}_x$}$
>
> | Parameters      | $z_x$ on $x$ | $z_{x}$ on ${x, t}$ | $z_x$ on $t$ | Mean |
> |-----------------|---------|--------------|---------|------------|
> | $w_1 = 0$        | 0.583   | 0.606        | 0.061   | 0.417      |
> | $w_2 = 0$         | 0.781   | 0.606        | 0.060   | 0.482      |
> | $w_3 = 0$         | 0.742   | 0.633        | 0.059   | 0.478      |
> | $w_1=w_2=w_3=1$      | 0.786 | 0.612    | 0.061 | **0.486**  |
>
> $\textbf{Table 2: Benchmark on $\mathbf{z}_{tx}$}$
> | Parameters      | $z_{tx}$ on $x$ | $z_{tx}$ on ${x, t}$ | $z_{tx}$ on $t$ | Mean |
> |-----------------|----------|---------------|----------|------------|
> | $w_1 = 0$          | 0.731    | 0.595         | 0.068    | **0.465**      |
> | $w_2 = 0$          | 0.725    | 0.556         | 0.037    | 0.439      |
> | $w_3 = 0$          | 0.748    | 0.575         | 0.066    | 0.463      |
> | $w_1=w_2=w_3=1$      | 0.732    | 0.594         | 0.068    | **0.465**      |
>
> $\textbf{Table 3: Benchmark on $\mathbf{z}_t$}$
> | Parameters      | $z_t$ on $x$ | $z_{t}$ on ${x, t}$ | $z_t$ on $t$ | Mean |
> |-----------------|---------|--------------|---------|------------|
> | $w_1 = 0$          | 0.333   | 0.389        | 0.069   | 0.264      |
> | $w_2 = 0$          | 0.334   | 0.372        | 0.066   | 0.257      |
> | $w_3 = 0$          | 0.331   | 0.401        | 0.081   | 0.271      |
> | $w_1=w_2=w_3=1$      | 0.334   | 0.393        | 0.088   | **0.272**      |
>
> When one of ${w_1, w_2, w_3}$ is set to 0, the remaining weights are set to 1.0. Based on this setup, we draw the following conclusions:
>
> The results of $z_x$ clustering on $x$, {$t, x$}, and $t$ indicate that
> - $w_1$ (the weight for $L_{sim}$) significantly enhances the performance of $z_x$ clustering on $x$ and $z_t$ clustering on $t$. This improvement is due to $L_{sim}$ increasing the similarity among $z_x$ and $z_t$.
> - $w_2$ (the weight for $L_{ct}$) enhances $z_{tx}$ clustering on {$t$, $x$} and $t$, as well as $z_t$ clustering.
> - $w_3$ (the weight for $L_{dist}$) improves $z_{tx}$ clustering on {$t$, $x$}and also enhances the separation within the spaces of $z_{tx}$, $z_t$, and $z_x$.
>
> It's important to note that when ${w_1, w_2, w_3} = {1,1,1}$, the performance of $z_{tx}$ clustering on {$t$, $x$}is not as strong as $z_x$ clustering on {$t$, $x$}. However, by increasing $w_1$, $w_2$, and $w_3$, the performance of $z_{tx}$ surpasses that of $z_x$ (see Appendix Figure 7).
> $\textbf{Necessary to use $z_{x}^0$ instead of $z_{x}$ in the function g, if $z_{x}$ is independent of $t$}$?:
>
> Thank you for raising this question. Yes, directly utilizing $z_x$ might achieve higher $R^2$ score, here we followed CPA, VCI and other methods and replace by $z_x^0$ to perform "counterfactual" prediction to further evaluate the learned disentangled representations.
>
> $\textbf{Assumption 4.3 is strong}$:
>
> Assumption 4.3 can be better understood through the lens of biological experimental design. If we aim to determine whether there is a covariate $x_1$-specific response to treatment $t_1$, we need to conduct two reference experiments: $(x_0, t_1)$ and $(x_0, t_0)$, where $x_0$ represents reference cells and $t_0$ represents control/reference treatments.
>
> By comparing the treatment effects between the pairs $\{(x_1, t_1), (x_1, t_0)\}$ and $\{(x_1, t_1), (x_0, t_1)\}$, we can assess whether the observed differences are significant and distinct (as described in Assumption 4.4). Without comparing $(x_1, t_1)$ and $(x_1, t_0)$, we cannot determine if $t_1$ has any effect on $x_1$. Similarly, without comparing $(x_1, t_1)$ and $(x_0, t_1)$, we cannot assess whether $t_1$ has a differential effect on $x_1$ than $x_0$. Only when both comparisons are made can we begin to analyze covariate-specific effects. This comparison allows us to confidently identify any $x_1$-specific response to $t_1$.
>
> This is a common approach in biological experimental design. For instance, an increasing number of experiments are being conducted for drug screening across various micro-environment organoid systems (covariates), as well as with different drugs, dosages, and time points [3]. The main goal is to discover these covariate-specific cellular responses.
>
>
> $\textbf{Assumptions empirically arguing that this is the case in common data sets}$:
>
> The substantive changes assumption (iii) means the different cell lines under the same drug treatment, the gene expression is significantly different (non-trivial). It can be seen that there are large parts of gene expression only influenced by cell types, etc.
> (iv) means under different treatments, the same cell (covariates) have significantly different responses to gene expression.
>
> These phenomena are all observed in biological literatures [1] [2] [3]. The Figures 3.B,C,D ($\textbf{Figure D in the rebuttal PDF file}$) in the Trellis paper [3] show evidence that these assumptions are common.
>
> $\textbf{Limiations}$: Due to the limited space, please check the response to $\textbf{Reviewer WECd}.$
>
> [1] Sanjay R. Srivatsan et al. ,Massively multiplex chemical transcriptomics at single-cell resolution.
> [2] McFarland, J.M., Paolella, B.R., Warren, A. et al. Multiplexed single-cell transcriptional response profiling to define cancer vulnerabilities and therapeutic mechanism of action.
> [3] Ramos Zapatero, María et al. Trellis tree-based analysis reveals stromal regulation of patient-derived organoid drug responses.

---

> > ### Comment · Reviewer_q35E · 2024-08-07
> >
> > Thank you for the extensive response. With additional information on hyperparameter tuning and ablation study, I have increased the score to weak accept.

---

> ### Author Response · Authors · 2024-08-08
>
> Thank you again for acknowledging our paper strengths and raising the score. We will further polish our manuscript and include the  hyperparameter tuning and ablation study in the final version.

---

### Official Review · Reviewer_yZVk · 2024-07-14

**Soundness:** 3
**Presentation:** 3
**Contribution:** 3
**Rating:** 6
**Confidence:** 4

**Summary:**

The authors present a novel method for causal representation learning in cellular perturbation settings, called Factorized Causal Representation, in which the authors learn disentangled latents for the cellular covariates, the treatment, and the interactions between them. They provide identifiability results following the methods of recent papers, and demonstrate how an implementation of the method, which uses a number of novel regularizers to enforce various causal constraints, achieves top results amongst comparable methods.

**Strengths:**

- novel causal representation learning method explicitly learning interactions between covariates and treatments
- novel regularizers for enforcing conditional independence between sets of variables
- achieves top cellular response prediction ($R^2$) over comparable conditions

**Weaknesses:**

- the datasets used have a relatively small number of similar treatments (16 at the most), so it is unclear how well the approach will work when applied to large diverse chemical libraries typical of real-life settings
- the decision not to evaluate on unseen treatments or cell contexts is surprisingly conservative, since the promise of causal models is in generalizing to unseen domains

**Questions:**

- why not evaluate on unseen treatments or cell contexts? Did you try and find the results to be comparatively poor?

---

> ### Author Rebuttal · Authors · 2024-08-06
>
> $\textbf{Dataset size}$:
>
> Due to the limited availability of public datasets and the high cost of single-cell sequencing, single-cell datasets with a large number of drugs are not yet available. However, as demonstrated by the theorems in our paper, incorporating more treatments with well-designed controls can enable the latent representations to capture more meaningful and disentangled information, thereby facilitating the algorithm's learning process. As more datasets become available in the future, we plan to conduct extensive benchmarks to evaluate our approach on a broader scale.
>
> $\textbf{Prediction for unseen perturbations}$ :
>
> Predicting responses to novel treatments is a pivotal and fast-evolving field in drug discovery. However, the biological literature indicates that cellular responses are highly context-dependent [1,2,3]. This complexity poses significant challenges for AI-driven drug discovery, which often struggles to achieve success in clinical trials [4].
>
> Our motivation for developing FCR stems from the need to understand how cellular systems react to treatments and identify conditions that can deepen our understanding of these responses. FCR enables the analysis of drug interactions with covariates and contextual variables. Additionally, predicting cellular responses to new treatments necessitates prior knowledge, such as chemical structure and molecular function, and comparisons with known treatments. Without this context, predictions can be unreliable.
>
> In this paper, our primary focus is not on predicting responses to unseen treatments. However, given the relevance of this topic, we conducted pilot experiments to showcase the potential future applications of FCR. The MultiTram-Plex and Multiplex-7 datasets share the same cell lines and Trametinib-24 hours treatment, along with other different treatments. By utilizing these two datasets, we established the following experimental settings for unseen prediction scenario:
>
>
> - **Drug Hold-out Setup**: We held out two cell lines, ILAM and SKMEL2, from the Multiplex-Tram dataset, which had been treated with Trametinib for 24 hours. We trained a FCR model using the remaining data, and this model is referred to as $M_h$. We denote the dataset (Multiplex-Tram dataset without Trametinib-24h treated ILAM and SKMEL2) as $D^h$
> - **Prior Knowledge Model**: We trained another FCR model, $M_p$, using the Multiplex-7 dataset, which includes the ILAM and SKMEL2 cell lines treated with Trametinib for 24 hours denoted as $D^p$. We treat this model, $M_p$, as a prior knowledge model.
> - **Transfer MLP**: We extract both $z_{tx}^{p}$ from $M_p$ and $z_{tx}^{h}$ from $M_h$ for $D^p$. Then we trained a 1-layer MLP (the same dimension as $z_{tx}^p$) to transfer $z_{tx}^p$ to $z_{tx}^h$, by minimize the MSE between them.
> - **Contextual Prior Representation**: We extracted the $z_{tx}^p$ representations from model $M_p$ for ILAM and SKMEL2 in holdout set. Then transfer $z_{tx}^p$ by the previous MLP to $\hat z_{tx}^h$ as a prior contextual embedding.  For the hold-out cell lines ILAM and SKMEL2 in the Multiplex-Tram dataset, we extracted  $z_{x}^h$ from model $M_h$, and Trametinib-24h $z_{t}^h$ from other treated cell lines in $D^h$
> - **Representation Matching**: Then we match the $\hat z_{tx}^h$ by $z_{x}^p$ similarity on ILAM and SKMEL2 across Multiplex-tram and Multiplex-7 in the prior knowledge model space.
> - **Prediction**: We predicted the unseen 24 hours Trametinib responses for the ILAM and SKMEL2 cell lines in holdout dataset using the formula:
> $\hat y =g(z_{x}^h,\hat z_{tx}^h, z_{t}^h)$,where $\hat z_{tx}^h$ is the corresponding matched prior contextual representation, $ z_t^h$ is the tramnib-24 average value in other cell lines.
> - **Evaluation**: We computed the $R^2$ and MSE for the top 20 differentially expressed genes (DEGs)  based on the predicted values and compared these results with those from the VCI model.The paper’s OOD prediction setups.)
> - $\textbf{The illustration of datasets and experiments setups can be found in the rebuttal PDF file, Figure A and B}$.
>
>
> We have the following results:
>
> $\textbf{Table: Unseen Prediction Results}$
> | Methods | $R^2$ | MSE  |
> |---------|-----|------|
> | FCR     |   $0.74\pm 0.03$  | $0.52 \pm 0.11$      |
> | VCI     |   $0.71 \pm 0.05$  |   $0.55 \pm 0.08$   |
>
> [1] Sanjay R. Srivatsan et al. ,Massively multiplex chemical transcriptomics at single-cell resolution.Science.
> [2] McFarland, J.M., Paolella, B.R., Warren, A. et al. Multiplexed single-cell transcriptional response profiling to define cancer vulnerabilities and therapeutic mechanism of action. Nat Commun.
> [3] Ramos Zapatero, María et al. “Trellis tree-based analysis reveals stromal regulation of patient-derived organoid drug responses.” Cell vol. 186,25 (2023).
> [4] Derek Lowe "AI drugs so far".

---

> ### Author Response · Authors · 2024-08-11
>
> Dear Reviewer **yZVk**:
>
> We sincerely appreciate your constructive suggestions on the unseen prediction. During the rebuttal stage, we worked diligently to design and execute the experiment to address your feedback (Please refer to the response above, as well as Figures A and B in the rebuttal FDF file). We would greatly value your feedback on the experiment and would be eager to discuss it further with you before the discussion period ends. Your insights will be crucial in helping us refine our paper.
>
> Thank you,
>
> All the authors

---

> > ### Comment · Reviewer_yZVk · 2024-08-12
> >
> > Thank you for addressing my review. The unseen cell line experiment is nice, and probably as good as can be done given the datasets. I've raised my score accordingly.

---

### Official Review · Reviewer_xvG5 · 2024-07-17

**Soundness:** 3
**Presentation:** 3
**Contribution:** 3
**Rating:** 6
**Confidence:** 3

**Summary:**

The paper presents a novel method, the Factorized Causal Representation (FCR), which leverages identifiable deep generative models to understand cellular responses to genetic and chemical perturbations across multiple cellular contexts. This method improves upon prior models by learning disentangled representations that delineate treatment-specific, covariate-specific, and interaction-specific factors, which are shown to be theoretically identifiable. The effectiveness of FCR is demonstrated on four single-cell datasets. The performance, measured by the R^2 score of conditional cellular responses, often marginally outperforms state-of-the-art methods, showcasing FCR’s potential in accurately modeling cellular dynamics.

**Strengths:**

- The paper successfully demonstrates the identifiability of treatment-specific, covariate-specific, and interaction-specific representations.
- The inclusion of the cell type classifier $f_{ct}$ and permutations discriminator $f_{dist}$ are innovative loss components.
- Sections 6.1 and 6.2 are well-designed to rigorously test the proposed method's ability to maintain the integrity of dimension reduction, decomposition, and conditional independence, validating the model's effectiveness in designing and training.

**Weaknesses:**

Choice of Evaluation Metric: A significant concern arises from the choice of the $R^2$ metric in section 6.3 for evaluating the performance of the Factorized Causal Representation (FCR) model. $R^2$ is suitable to assess whether a model adequately explains the variation in the response variable. However, it is not necessarily an indicator of the model's predictive accuracy.  Moreover, the near-identical $R^2$ reported for FCR, VCI, CPA, and sVAE (as shown in Table 1) suggest that $R^2$ gives limited insights into the comparative effectiveness of these methods.

There are some minor misformats, which I noticed while reading.
- Lines 35, 54: citation format issue
- Line 294: subscript format issue

**Questions:**

Can you elaborate more about the design choice of the evaluating metric?

**Limitations:**

- Generalizability: The paper does not show the method’s applicability to apply on new dataset. This limitation raises concerns about how well the FCR can adapt to other biological datasets or experimental setups that deviate from the four testing datasets, which restricts the method’s utility.
- Interpretability: The model lacks mechanisms for clear interpretation, particularly in linking gene expression profile changes directly to specific treatments. So the decomposition does not provide insights into how gene expression is affected by specific treatments.

I understand that both limitations are significantly challenging and it is unrealistic to tackle both of them in one paper while they highlight critical areas for future research.

---

> ### Author Rebuttal · Authors · 2024-08-06
>
> $\textbf{Evalution metrics}$:
> Please check the **General Response to All Reviewers a**.
>
> For the performance of $R^2$, we assessed the statistical significance of those results between FCR and second best methods using an paired t-test and showed the p-values with the corresponding sample size of the test set.
> $\textbf{Table : p-value for $R^2$ performance}$
> | Dataset         | p-value (# cells) |
> |-----------------|-------------------|
> | sciPlex         | 0.0006 (295)      |
> | Multiplex-Tram  | 0.0056 (266)      |
> | MultiPlex-7     | 0.0001 (318)      |
>
> The results indicate that FCR statistically significantly out-performed the baselines across datasets on sci-Plex, multiPlex-Tram and multiPlex-7.
>
> We will incorporate the discussion of evaluation and t-test results in our final manuscript.
>
> $\textbf{Generalizability}$:
>
> We concur with the reviewer that generalizability is a very important topic in the computational biology field. Given the high cost of data collection, we are still limited to test and evaluate our models on the few public large scale single cell drug screening datasets that are available. As availability improves, there will be improved possibilities for benchmarking, especially in the out-of-distribution scenario.
>
> $\textbf{Interpretability}$:
>
> There are generally two approaches to interpreting latent representations, which we also plan to explore in future work:
> 1. **Latent Traversal**: Since $z_{tx}$ is element-wise identifiable, we can use latent traversal across all dimensions of $z_{tx}$. This involves fixing all latent variables $z_{tx}^i$ to their inferred values for each cell's gene expression and then varying the value of one latent at a time to observe the reconstructed gene expression. This method allows us to see how each dimension of $z_{tx}$ influences gene expression levels.
> 2. **Interpretable Decoder System**: We can also employ an interpretable decoder, such as LDVAE [1], which uses a linear decoder. By analyzing the weights of the latent representations assigned to each gene, we can interpret how different representations influence each gene.
> 3. **On-going Work**: In the light of the neural additive model [2], we are developing a non-linear interpretable encoder as an extension of LDVAE. Specifically, for each latent dimension, there is a sub-neural network to capture information specific to each latent dimension. A logistic regression layer is added to the end of the neural network. So, by analyzing the logistic regression weights (loading matrix), we can conclude how much one latent variable contributes to the reconstruction of a specific gene. **An illustration of this model can be found in the rebuttal PDF file Figure D**.
>
> [1]Svensson, V., et al.(2020). Interpretable factor models of single-cell RNA-seq via variational autoencoders. Bioinformatics (Oxford, England), 36(11), 3418–3421.
> [2]Agarwal, R., et al. (2021). Neural additive models: Interpretable machine learning with neural nets. Advances in neural information processing systems, 34, 4699-4711.

---

### Official Review · Reviewer_vnnM · 2024-07-17

**Soundness:** 3
**Presentation:** 3
**Contribution:** 3
**Rating:** 6
**Confidence:** 4

**Summary:**

The authors develop a modification of the identifiable nonlinear ICA algorithm of Khemakhem et al. 2020, suited for scenarios with two disjoint groups of auxilliary variables x and t and their interactions, with an application of treatment effect on cell gene expression.
The authors propose a learning objective designed to enforce the conditional independence assumptions of the model, and evaluate it against baselines from the disentanglement, ICA, and cellular response estimation.

**Strengths:**

- the proposed method is justified theoretically
- the paper is written in a clear and concise manner
- extensive selection of baselines

**Weaknesses:**

- I fail to see the reasoning of comparing the conditional independence results (6.3) with the other baselines, which were not designed to have separate latent subspaces for x,t, and tx, and thus it does not seem surprising that the conditional independence does not hold for random subsets of them?

- Results of the cellular response tasks are barely better than the baselines - did the authors conduct any tests to see if the differences are statistically significant?

**Questions:**

- Equation (6) (the constraint for interaction between x and t) - how does that guarantee the interactions? The learned embeddings can have empty dimensions, e.g, k(x) = [k_1(x), …, k_n(x), 1, …, 1] and k(t) = [1, …, 1, k_1(t), …, k_m(t)] so that the Hadamard product becomes [k(x), k(t)]
- Equation (7) - shouldn’t f_dis also take x as an argument since you want to enforce a conditional indpendence? Or do you train a separate f_dis for each possible value of x that you condition on (probably not feasible)?
- If the model is supposed to be fully identifiable wrt. z_xt, then I would suggest evaluating this with the appropriate metrics on synthetically generated data in addition to the clustering objective (e.g., MCC  as in Khemakhem et al. 2020, or any of the disentanglement metrics [Locatello et al. 2019])
- How were the hyperparameters selected for the cellular response task?

**Limitations:**

- the algorithm has 3 hyperparameters, which might make training difficult. Details on hyperparameter selection were ommited
- the evaluation is limited - results of the downstream task of predicting cellular response yield limited improvements, and the identifiability of  z_{t,x} is not properly evaluated (with e.g., the MCC metric using synthetic data)

---

> ### Author Rebuttal · Authors · 2024-08-06
>
> $\textbf{Comparing the conditional independence results (6.3) with the other baselines}$:
>
>
> We thank the reviewer for raising this important question.  I would like to clarify the experimental setup. The goal of this experiment is to demonstrate that FCR is capable of learning conditionally independent spaces for $z_x$, $z_t$, and $z_{tx}$. Our algorithm is the first to specifically address the $x$, $t$, and $tx$ subspaces, which makes it challenging to find direct baselines for comparison. However, the baselines such as **iVAE**, **betaVAE**, **factorVAE**, and **sVAE** are semi-supervised or unsupervised disentangled representation learning methods, they aim to learn representations that should be independent or conditionally independent given $x$ and $t$. Additionally, methods like **CPA** and **VCI** enforce certain invariance relations with respect to $t$ and $x$, potentially leading to some level of conditional independence in the learned representations.
>
> However, the challenge in making this comparison lies in not knowing which blocks of latent variables induced by baseline methods satisfy the specific conditional independence relations of interest. To ensure a fair comparison, we exhausted different combinations of the representations for the tests and reported the best results. This comparison is intended to demonstrate that **FCR** can effectively learn more meaningful and nuanced conditional independence relations in $x$, $t$ and $tx$, compared to the baselines. We will ensure that this clarification is included in the manuscript.
>
> $\textbf{Differences are statistically significant}$:
>
> Thank you for raising this concern. We tested for significance of changes in mean between the results from the FCR method and those of the second-best performing method (paired t-test). The p-values, along with the corresponding cell number in the testing set appears in the table below.
> #### Table: p-value for $R^2$ performance
> | Dataset         | p-value (# cells) |
> |-----------------|-------------------|
> | sciPlex         | 0.0006 (295)      |
> | Multiplex-Tram  | 0.0056 (266)      |
> | MultiPlex-7     | 0.0001 (318)      |
>
> The results indicate that FCR significantly out-performed the baselines across cell lines on sci-Plex, multiPlex-Tram and multiPlex-7. On MultiPlex-9, where FCR is slightly outperformed by CPA, the differences are not significant. We will make sure to include all p-values in the updated manuscript.
>
> To propose an additional metric beyond $R^2$, we also evaluated the mean square error (MSE), for the top 20 Differentially Expressed genes (DEGs). These 20 genes are  measured on the largest difference for each cell line with drug treatments compared to control samples (following [1]). Note here, we didn’t compare with CINEMA-OT and scGEN because they are only for binary treatments (case-control data). For detailed results, please check the results in **General Response to ALL Reviewers a**.
> Generally speaking, FCR outperforms other baselines in 3 of 4 datasets.
>
> $\textbf{Concerns on Equation (6) (the constraint for interaction between $\mathbf{x}$ and $\mathbf{t}$)}$:
>
> The reviewer asked the question "How does that guarantee the interactions? The learned embeddings can have empty dimensions, e.g, $k(x) = [k_1(x), \ldots, k_n(x), 1, \ldots, 1]$ and $k(t) = [1, \ldots, 1, k_1(t), \ldots, k_m(t)]$ so that the Hadamard product becomes $[k(x), k(t)]$".
>
> We apologize for this misleading language. Essentially, we did not want to claim any theoretical guarantee in this case, but instead motivate the specific architecture choice. We therefore changed the language to "in order to encourage the prior for $z_{xt}$ to capture interactions between $x$ and $t$, we design the functions $f^p_{\mu}$ and $f^p_{\Sigma}$  to be of the form $k(x) \otimes k(t)$, where $\otimes$ denotes the Hadamard product.
>
> $\textbf{x as an argument for $f_{dis}$}$:
> We agree with the reviewer that it would be much clearer to put $x$ as an argument of  $f_{dist}$.  It is already the case that the function $f_{dist}$ implicitly considers $x$ as an argument because we perform a random permutation of the triplet $(z_x, z_{tx}, x)$ under the condition of the same $x$ (i.e., the permutation or shuffle occurs only within a set of $z_x$ and $z_{tx}$ under the same $x$). As a result, the input to $f_{dist}$ is the set of newly generated ($z_x$, $\hat z_{tx}$), where we ensure that $z_x$ and $\hat z_{tx}$ correspond to the same $x$. The outputs of $f_{dist}$ simply are the labels indicating whether a permutation occurred or not. Therefore, we can say that $f_{dist}$ effectively takes $x$ into account. We briefly discuss this process in lines 226-230. We will make sure to update it in our final manuscript.
>
> $\textbf{Simulation studies on synthetic data}$:
> Please check the **General Response to ALL Reviewers c**.
>
>
> $\textbf{Hyperparameter selection}$:
> Thank you for raising this important question, we appreciate that it is a critical point for evaluating how well the method works. Please check **General Response to All Reviewers b**.

---

> > ### Comment · Reviewer_vnnM · 2024-08-08
> >
> > Thank you for your answers.
> >
> > While most of my questions were clarified, I would suggest a different choice for the additional metric instead of MSE, e.g., the rank correlation mentioned by authors in the response. Unless I am missing something MSE does not bring too much new insight beyond R2, as they are both measuring the squared residuals, with the main difference being that R2 is assuming standardized targets.
> >
> > Nevertheless in light of the other answers I raised my score accordingly

---

> ### Author Response · Authors · 2024-08-10
>
> Thank you very much for raising the score. For the MSE evaluation, we selected the top 20 differentially expressed genes, identified by conducting a comparison between treated and untreated cells, and ranked them based on t-test p-values, we considered the rank information to some extent in this evaluation.  Additionally, we plan to add the Spearman correlation comparison in the final version (The table below shows FCR and VCI results). Again, thank you for your valuable comments which helped us improve our work a lot!
> **Table: Spearman Correlation (top 500 highly variable genes)**:
> | Datasets         | FCR         | VCI         |
> |------------------|-------------|-------------|
> | sciPlex          | **0.84(0.07)** | 0.82(0.08) |
> | multiplex-Tram   | **0.87(0.05)**  | 0.85(0.06)  |

---

### Author Rebuttal · Authors · 2024-08-06

$\textbf{General Response to ALL Reviewers}$:

We sincerely thank all the reviewers for your thorough evaluation of our paper and your recognition of our contributions. We want to address some of the common questions and update new experiment results here, and we will have detailed responses for each reviewer below:

$\textbf{a. Evaluation Metrics}$:

The choice of evaluation metrics for the drug response prediction task is indeed a topic of much discussion. We agree that $R^2$ is well-suited to assess how well a model explains the variation in the response variable.

We chose $R^2$ as our evaluation metric for two main reasons:
- Previous works, including **VCI**, **CPA**, and **sVAE**, have also used $R^2$ as a standard evaluation metric, providing a basis for comparison.
- Single-cell transcriptomics data is often noisy, sparse, may contain batch effects, and highly related to library size making direct evaluation of gene prediction accuracy challenging. Given the noisiness of this data, researchers commonly use relative gene enrichment analysis to compare different cell types.

Other evaluation metrics could be:
- the Mean Squared Error (**MSE**) of the top differentially expressed genes (**DEGs**) for post-treatment [1].
- the Spearman correlation between predicted genes and ground truth across all the cells.
- the cosine similarities between the predicted genes and ground truths.

We added MSE results for the top 20 DEGs. These 20 genes are selected for showing statistically significant differences in expression levels for each cell line with drug treatments compared to control samples. The same procedures are also carried out in [1]. Note here, that we didn’t compare with CINEMA-OT and scGEN because they are only for binary treatments.

$\textbf{Table 1: MSE (Top 20 DEGs)}$

| Datasets        | FCR (ours)         | VCI         | CPA         | sVAE        |
|-----------------|-------------|-------------|-------------|-------------|
| sciPlex         | **0.12 (0.03)** | 0.15 (0.04) | 0.15 (0.04)  | 0.16 (0.05)  |
| multiPlex-tram  | **0.10 (0.07)** | 0.13 (0.08) | 0.13 (0.08)  | 0.14 (0.07)  |
| multiPlex-7     | **0.18 (0.07)** | 0.21 (0.08) | 0.22 (0.08)  | 0.23 (0.07)  |
| multiPlex-9     | **0.24 (0.06)** | 0.27 (0.04) | **0.23 (0.06)** | 0.26 (0.05)  |

FCR still outperforms the competing methods in three datasets while CPA achieves slightly better results on multiPlex-9.

$\textbf{b. Hyperparameter Selection}$:

We split the data into four datasets: train/validation/test/prediction, following the setup from previous works [2]. First we hold out the 20% of the control cells for the final cellular prediction tasks (pred). Second, we hold 20% of the rest of data for the task of clustering and statistical test (test). Third, the data excluding the prediction and clustering/test sets are split into training and validation sets with a four-to-one ratio.

For the hyperparameter tuning procedure, conduct the exhaustive hyperparameter grid search with n_epoch=100 on the loss assessed on the validation data. The hyperparameter search space is

$w_1 = [0.5, 1 ,2, 3, 4, 5, 6, 7, 8, 9, 10]$
$w_2 = [0.5, 1 ,2, 3, 4, 5, 6, 7, 8, 9, 10]$
$w_3 = [0.1, 0.3, 0.5, 0.7, 0.9, 1, 3, 5, 7, 10]$

We will make sure to include this in our final manuscript.

$\textbf{c. Simulation Study}$:

To address **Reviewer vnnM**'s question on synthetic data experiments to demonstrate the identifiability of $z_{tx}$.

By following the [3,4] protocol of simulation data. For the simplicity of simulation,  we set the dimension of $z_x$ and $z_t$ equal to 1, and the $z_{xt}$ dimension as 4. We have the following setups. For the simplicity of the simulation, we output the $y$ (dimension is 96) as real numbers instead of count data with a sample number of 5000.

$t \sim \textrm{Unif}([1,2,3])$,
$x \sim \textrm{Unif}([100, 1000, 5000])$,
$z_x \sim \textrm{Normal}(x/2, 1)$,
$z_t \sim \textrm{Normal}(t/2,1)$,
$z_{tx} \sim \textrm{Normal}(x*t, I_4)$,
$g$ is a 2-layer MLP with the Leaky-ReLU activation function [3].
To measure the component-wise identifiability of the changing components, we compute the Mean Correlation Coefficient (MCC) between $z_{tx}$ and $\hat z_{tx}$ .  A higher MCC indicates a higher extent of identifiability, and MCC reaches 1 when latent variables are perfectly component-wise identifiable.  We computed the mean correlation coefficient (MCC) between the original sources and the corresponding latents sampled from the learned posterior. We followed iVAE [3], and first calculated all pairs of correlation coefficients between source and latent components. We then solve a linear sum assignment problem to assign each latent component to the source component that best correlates with it, thus reversing any permutations in the latent space. The UMAP projections of the ground truth $z_{tx}$ and estimated $\hat z_{tx}$ which showed perfect separation are shown in Figure C of the PDF file.

$ \textbf{Table 2 : Simulation Results on MCC}$
| Methods    | MCC  |
|------------|------|
| FCR        | **0.91 (0.03)**  |
| beta-VAE   |  0.38 (0.12)    |
| iVAE       |    0.77 (0.07)  |
| factor-VAE |    0.37 (0.08)  |

$\textbf{d. Unseen Prediction}$:
For the unseen prediction experiment setup and results, please check the response to **Reviewer yZVk**.

$\textbf{Note: We have new figures in the attached PDF file}$

$\textbf {References}$:

[1] Roohani, Y., et al. (2024). Predicting transcriptional outcomes of novel multigene perturbations with GEARS. Nature Biotechnology.
[2] Lotfollahi M, et al. Predicting cellular responses to complex perturbations in high-throughput screens. Mol Syst Biol.
[3] Khemakhem, Ilyes, et al. "Variational autoencoders and nonlinear ica: A unifying framework." International conference on artificial intelligence and statistics. PMLR, 2020.
[4] Lopez, Romain, et al. "Toward the Identifiability of Comparative Deep Generative Models." Causal Learning and Reasoning. PMLR, 2024.

---

### Decision · Program_Chairs · 2024-09-25

**Decision:**

Accept (poster)

**Comment:**

The method proposes a method for learning disentangled causal representations of covariates, treatment and their interaction
The learning objective enforces the conditional independence assumptions and is theoretically well-justified.
Performance is tested in and extensive evaluation showing that the method performs significantly better than a large set of baselines. The paper is well-written and tackles a relevant and hard problem. The reviewers criticise minor issues like complexity of the model and choices in the evaluation scheme.

All five reviewers vote towards acceptance.